# Efficient Adaptation of Large Vision Transformer via Adapter Re-Composing

**Wei Dong**[1,4]     **Dawei Yan**[1]     **Zhijun Lin**[2]     **Peng Wang**[3] *

[1]College of Information and Control Engineering,
Xi'an University of Architecture and Technology.
[2]School of Computer Science, Northwestern Polytechnical University.
[3]School of Computer Science and Engineering,
University of Electronic Science and Technology of China.
[4]Xi'an Hypersonic Measurement Technology Co., Ltd.

## Abstract

The advent of high-capacity pre-trained models has revolutionized problem-solving in computer vision, shifting the focus from training task-specific models to adapting pre-trained models. Consequently, effectively adapting large pre-trained models to downstream tasks in an efficient manner has become a prominent research area. Existing solutions primarily concentrate on designing lightweight adapters and their interaction with pre-trained models, with the goal of minimizing the number of parameters requiring updates. In this study, we propose a novel Adapter Re-Composing (ARC) strategy that addresses efficient pre-trained model adaptation from a fresh perspective. Our approach considers the reusability of adaptation parameters and introduces a parameter-sharing scheme. Specifically, we leverage symmetric down-/up-projections to construct bottleneck operations, which are shared across layers. By learning low-dimensional re-scaling coefficients, we can effectively re-compose layer-adaptive adapters. This parameter-sharing strategy in adapter design allows us to further reduce the number of new parameters while maintaining satisfactory performance, thereby offering a promising approach to compress the adaptation cost. We conduct experiments on 24 downstream image classification tasks using various Vision Transformer variants to evaluate our method. The results demonstrate that our approach achieves compelling transfer learning performance with a reduced parameter count. Our code is available at https://github.com/DavidYanAnDe/ARC.

## 1 Introduction

The utilization of large-scale pre-trained models for various downstream tasks has garnered significant interest in the computer vision community [1; 2; 3; 4]. These models continually push the boundaries of downstream task performance while eliminating the need for task-specific model design and training. In early attempts, a commonly adopted transfer learning strategy involved directly fine-tuning a pre-trained model on downstream tasks. However, the full fine-tuning strategy suffers from two major drawbacks: (1) Updating large-scale parameters is prohibitively expensive and typically requires a substantial amount of training data to prevent overfitting. (2) As the sizes of state-of-the-art pre-trained models continue to increase, it becomes impractical and unsustainable to store a distinct set of model weights for each downstream task.

---

*Corresponding author. Email address: `p.wang6@hotmail.com`

37th Conference on Neural Information Processing Systems (NeurIPS 2023).

In contrast to fine-tuning the entire pre-trained network, recent research has focused on parameter-efficient model adaptation. The core idea behind this line of work is to keep the majority of pre-trained parameters frozen and only update or introduce a small fraction of task-specific parameters. Several methods fall under this umbrella, including prompt tuning [5; 6], visual adapter [7; 8], and linear feature modulation [9]. These methods have demonstrated competitive or even superior performance compared to full fine-tuning while significantly reducing the adaptation cost. They differ in the design of lightweight adapters and how these adapters interact with the pre-trained parameters. Prompt tuning methods [5; 6] adapt the features of pre-trained Vision Transformers by introducing trainable task-specific tokens into one or multiple attention layers. Visual adapter [7; 8] injects a non-linear lightweight adapter with a bottleneck architecture between layers of the pre-trained model to adjust the feature distribution. Such non-linear adapters are simplified using linear transformations such as shifting and scaling [9] to directly modulate the pre-trained features.

In this work, we not only focus on designing lightweight adapters but also emphasize the importance of adaptation parameter reusability in further compressing the adaptation cost. We adopt a low-rank design for the adapter using a bottleneck operation but propose a novel approach. Unlike other methods that place the adapter in different layers and directly learn different parameters for each adapter to cater to layer-wise variation, we propose sharing the down/up projections in the low-rank adapter across different layers and simply learning low-dimensional re-scaling coefficients to re-compose the linear projections into layer-adaptive adapters. The idea of Adapter Re-Composing (ARC) is motivated by the observation that naturally derived adaptation matrices can exhibit extremely low-rank characteristics, even when not explicitly designed as such. This implies the possibility of using a shared "basis" to re-compose the adapters. Furthermore, we design the low-rank adapters using symmetric down-projection and up-projection matrices, which further reduces the parameter size. Due to their linear nature and careful positioning design, our adapters can be seamlessly integrated into the pre-trained network, as in [7; 9; 10], without adding extra computation during the inference phase.

We evaluate our method on various large Vision Transformer models, including ViT-B [1] and its variants such as ViT-L [1], ViT-H [1], and Swin-B [11], using 24 downstream image classification benchmark datasets. The experimental results demonstrate that our method achieves compelling transfer learning performance while maintaining a smaller parameter size.

The key contributions of this paper are summarized as follows:

- We approach efficient pre-trained model adaptation from a novel perspective by exploring the reusability of adaptation parameters, which goes beyond existing works that primarily focus on the lightweight design of adapter structures.

- We introduce the Adapter Re-Composing (ARC) strategy, which shares bottleneck operation's down-/up-projections across layers and utilizes lower-dimensional re-composing coefficients to create layer-adaptive adapters. This approach enables fewer parameters than prior works.

- Our parameter sharing scheme in the ARC method prevents a linear increase in parameter size with the number of layers, ensuring better scalability, particularly for larger-scale models.

- Through extensive experiments on various Vision Transformer variations and numerous downstream tasks, we show that our method achieves highly competitive transfer learning performance while maintaining a relatively low level of additional parameter.

## 2  Related work

In this section, we present a concise review of the existing literature, focusing on two key areas: (1) Pre-training and fine-tuning, and (2) Parameter-efficient transfer learning.

**Pre-training and fine-tuning.**    Pre-training and fine-tuning, also known as transfer learning [12; 13; 14], is a popular approach that utilizes large-scale datasets [15; 16; 17; 18; 19] to train models for adaptation to different downstream tasks. By extracting knowledge from these datasets and encoding it into parameters, models can be fine-tuned for specific tasks, resulting in improved performance compared to models without pre-training. The availability of large-scale datasets [17] has led to

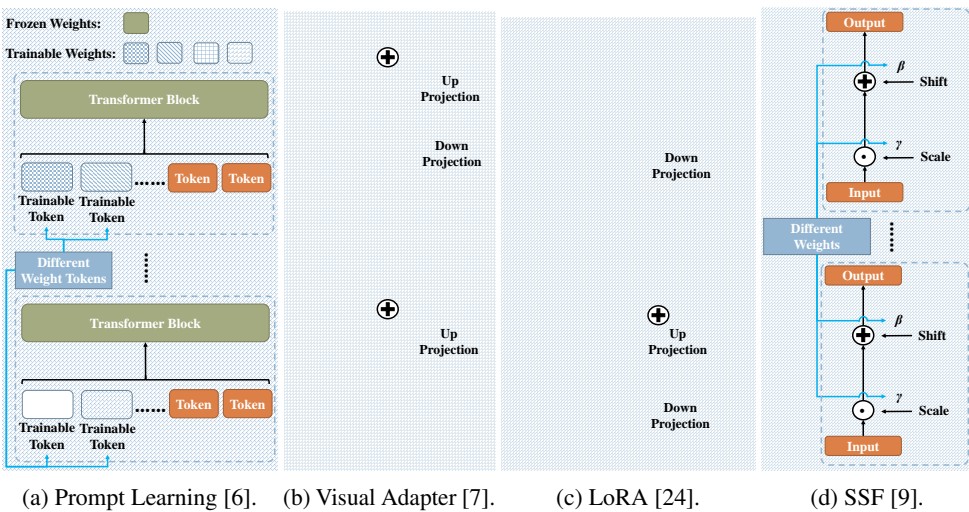

(a) Prompt Learning [6]. (b) Visual Adapter [7]. (c) LoRA [24]. (d) SSF [9].

Figure 1: Visual summary of typical parameter-efficient pre-trained model adaptation methods.

significant advancements in performance and convergence speed for downstream tasks. Scaling up models [20; 21] to handle the growing data volume enhances data utilization and improves the efficiency and robustness of pre-trained models across various data distributions and noise levels. Large models [1; 11; 22] also benefit from self-supervised pre-training [2; 23] using unlabeled data, reducing the cost, duration, and quality issues associated with human annotation. By leveraging this approach, models can effectively extract knowledge from readily available unlabeled data, further enhancing their generalization performance in downstream tasks.

**Parameter-efficient transfer learning.** The exponential increase in model parameters presents a computational challenge when fine-tuning the entire network on downstream tasks. In the field of NLP, researchers have explored parameter-efficient transfer learning approaches [7; 24; 25; 26; 27; 28] that train a subset of the model or add new modules with fewer parameters while achieving comparable or even superior performance. Inspired by the success of NLP, several notable works [8; 29; 30] have emerged in the computer vision domain. One approach, known as Prompt Tuning [5; 6], addresses the distribution mismatch between pre-training and downstream tasks by learning task-specific tokens. Adapter-like methods [8; 7; 29] insert trainable modules, such as MLPs with activation functions and residual structures, into the network to facilitate transfer learning. LoRA [24] exploits the low-rank update to a large-scale frozen model and introduces a bypass to the original parameter matrix to mimic the fine-tuning of the entire model parameters. SSF [9] introduces lightweight scaling and shift operations to modulate the pre-trained representations. The core concepts of these aforementioned works are visually summarized in Fig. 1.

As an advancement in visual adaptation, ARC addresses the limitations of Prompt Tuning, which requires different prompt designs for different downstream tasks. Moreover, ARC introduces the innovative concept of bottleneck matrix reuse, achieving state-of-the-art performance with minimal adaptation cost comparing to other rank-decomposition strategies. Additionally, ARC employs a linear design for the adapters and inherits the benefits of re-parameterization [9; 10; 24], ensuring that inference does not introduce any additional computational complexity.

## 3 Approach

In this section, we start by providing an introduction to the notations, symbols, and background related to Vision Transformers, which is followed by the presentation of our proposed Adapter Re-Composing (ARC) method. ARC focuses on efficient transfer learning for Vision Transformers by reusing rank-decomposition projections to adaptively compose layer-wise adapters, thereby reducing the size of learnable parameters. Additionally, we discuss the insights gained from our architecture design.

### 3.1 Preliminary

A plain Vision Transformer model contains a patch embedding layer and multiple encoder layers. Given an input image $\mathbf{X} \in \mathbb{R}^{H \times W \times C}$, the patch embedding layer first splits the image into a sequence of flattened patches $\mathbf{X}_{\text{patches}} \in \mathbb{R}^{N \times (P^2 \cdot C)}$, where $(H, W)$ is the resolution of the input image, $(P, P)$ is the resolution of each patch, $C$ denotes the number of input channels, and $N = H \cdot W / P^2$ is the number of tokens. Subsequently, the image patches are mapped to a $D$-dimensional embedding space through a linear projection $\mathbf{W} \in \mathbb{R}^{(P^2 \cdot C) \times D}$. A learnable [class] token vector $\vec{\boldsymbol{x}}_{\text{cls}} \in \mathbb{R}^D$ is then prepended to the sequence of $\mathbf{X}_{\text{patches}}$, and the position embeddings $\mathbf{X}_{\text{pos}} \in \mathbb{R}^{(N+1) \times D}$ are added to the sequence. The output of the patch embedding layer can be expressed as follows:

$$\mathbf{X}_{\text{emb}} = [\vec{\boldsymbol{x}}_{\text{cls}}^{\text{T}}; \mathbf{X}_{\text{patches}} \mathbf{W}] + \mathbf{X}_{\text{pos}}, \tag{1}$$

where $[\cdot; \cdot]$ denotes concatenation operation. This output is then fed into several consecutive encoder layers, each consisting of a Multi-Head Attention (MHA) block and a Feed-Forward Network (FFN) block. LayerNorm (LN) is applied before each block, and residual connections are applied thereafter. The process of $l$-th encoder layer is defined as:

$$
\begin{aligned}
\mathbf{X}^{(l)\prime} &= \text{MHA}(\text{LN}(\mathbf{X}^{(l-1)})) + \mathbf{X}^{(l-1)}, \\
\mathbf{X}^{(l)} &= \text{FFN}(\text{LN}(\mathbf{X}^{(l)\prime})) + \mathbf{X}^{(l)\prime},
\end{aligned}
\tag{2}
$$

where $\mathbf{X}^{(l-1)}$ denotes input tokens in $l$-th layer, $\mathbf{X}^{(l)\prime}$ indicates intermediate representations produced by MHA, and the output of $l$-th layer is $\mathbf{X}^{(l)}$.

In MHA block, each Attention Head (AH) module utilizes the weight matrices $\mathbf{W}_q^{(l)} \in \mathbb{R}^{D^{(l-1)} \times D_h^{(l)}}$, $\mathbf{W}_k^{(l)} \in \mathbb{R}^{D^{(l-1)} \times D_h^{(l)}}$, and $\mathbf{W}_v^{(l)} \in \mathbb{R}^{D^{(l-1)} \times D_h^{(l)}}$ for the *query*, *key*, and *value* operations, respectively. These operations enable an exclusive attention mechanism on the normalized feature representations $\mathbf{X}_{\text{norm}}^{(l-1)} = \text{LN}(\mathbf{X}^{(l-1)})$:

$$\mathbf{X}_h^{(l)\prime} = \text{AH}_h(\mathbf{X}_{\text{norm}}^{(l-1)}) = \text{softmax}\left(\frac{(\mathbf{X}_{\text{norm}}^{(l-1)} \mathbf{W}_q^{(l)})(\mathbf{X}_{\text{norm}}^{(l-1)} \mathbf{W}_k^{(l)})^{\text{T}}}{\sqrt{D_h^{(l)}}}\right)\mathbf{X}_{\text{norm}}^{(l-1)} \mathbf{W}_v^{(l)}, \tag{3}$$

where $D_h^{(l)} = \frac{D^{(l)}}{M}$ is the feature dimensionality of the output representations $\mathbf{X}_h^{(l)\prime}$ for each attention head and $M$ represents the number of attention heads. The MHA block concatenates multiple $\{\mathbf{X}_h^{(l)\prime}\}$ in sequence and generates the outputs through a linear projection $\mathbf{W}_o^{(l)} \in \mathbb{R}^{(M \cdot D_h^{(l)}) \times D^{(l)}}$:

$$\mathbf{X}^{(l)\prime} = \text{MHA}(\mathbf{X}_{\text{norm}}^{(l-1)}) = [\text{AH}_1(\mathbf{X}_{\text{norm}}^{(l-1)}), \cdots, \text{AH}_M(\mathbf{X}_{\text{norm}}^{(l-1)})]\mathbf{W}_o^{(l)}. \tag{4}$$

The FFN block consists of two linear projections with the GELU activation function in between:

$$\mathbf{X}^{(l)} = \text{FFN}(\mathbf{X}_{\text{norm}}^{(l)\prime}) = \text{GELU}(\mathbf{X}_{\text{norm}}^{(l)\prime} \mathbf{W}_1^{(l)})\mathbf{W}_2^{(l)}, \tag{5}$$

where $\mathbf{W}_1^{(l)} \in \mathbb{R}^{D^{(l)} \times 4 \cdot D^{(l)}}$ and $\mathbf{W}_2^{(l)} \in \mathbb{R}^{4 \cdot D^{(l)} \times D^{(l)}}$ denote two linear projection matrices, and $\mathbf{X}_{\text{norm}}^{(l)\prime} = \text{LN}(\mathbf{X}^{(l)\prime})$.

### 3.2 Adapter Re-Composing method

We observe that the majority of existing adaptation methods introduce adapters into various layers and learn separate parameters for adapting the pre-trained model to specific downstream tasks. Previous studies [8; 24] have shown the effectiveness of leveraging the low-rank properties of adapters to fine-tune frozen pre-trained models. Inspired by these findings, we propose a novel approach to create a unified linear space across different adapters to enhance parameter efficiency and adaptation performance.

**Architecture.** The ARC architecture incorporates a bottleneck operation for adapters, which consists of three key components: a linear down-projection, layer-specific re-scaling coefficients, and a linear up-projection. This architecture is illustrated in Fig. 2. To facilitate the reusability of adaptation matrices, we have developed a sharing and re-composing scheme for the ARC method.

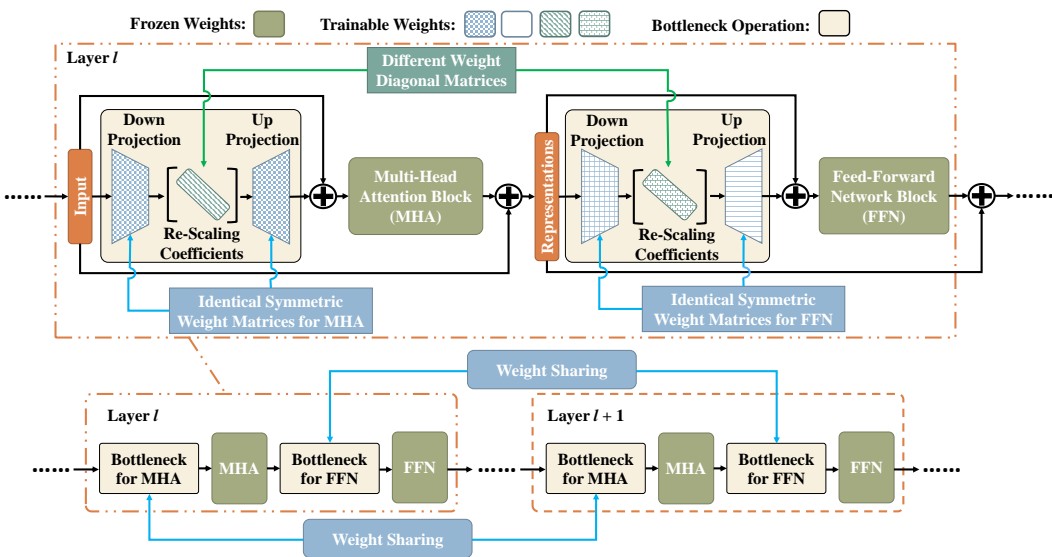

Figure 2: Illustration of the proposed Adapter Re-Composing Method.

This scheme involves sharing linear projections across layers and learning low-dimensional re-scaling coefficients to re-compose the layer-adaptive adapters. In addition, to enhance parameter compression, we leverage symmetric down-projection and up-projection in a single bottleneck operation:

$$\mathbf{W}_{\text{up}} = (\mathbf{W}_{\text{down}})^{\text{T}}, \tag{6}$$

where $\mathbf{W}_{\text{down}} \in \mathbb{R}^{D \times D'}$ and $\mathbf{W}_{\text{up}} \in \mathbb{R}^{D' \times D}$ with $D' \ll D$ denote shared down-projection and up-projection matrices across different layers; $D'$ represents the hidden dimensionality of the projections. To accommodate the variations across different layers, we learn re-scaling coefficients to re-compose the layer-adaptive adaptation matrices. These coefficients are then diagonalized into a diagonal matrix $\mathbf{C}^{(l)} \in \mathbb{R}^{D' \times D'}$ specific to each layer $l$. This diagonal matrix allows for efficient and effective adjustment of the adaptation parameters at each layer. Formally, given an input $\mathbf{X}_{\text{in}} \in \mathbb{R}^{(N+1) \times D}$, the output of our ARC module is:

$$\mathbf{X}_{\text{out}} = \text{ARC}(\mathbf{X}_{\text{in}}) = \mathbf{X}_{\text{in}} \mathbf{W}_{\text{down}} \mathbf{C}^{(l)} \mathbf{W}_{\text{up}} + \mathbf{X}_{\text{in}}. \tag{7}$$

Unless otherwise specified, we adhere to the default configuration of inserting our ARC modules sequentially before both the MHA and FFN blocks in the Vision Transformer. The influence of adapter positions will be discussed in Section 4.3. Therefore, the Vision Transformer incorporating our ARC modules can be formulated as follows:

$$\mathbf{X}^{(l)\prime} = \text{MHA}(\text{ARC}_{\text{MHA}}(\text{LN}(\mathbf{X}^{(l-1)}))) + \mathbf{X}^{(l-1)},$$
$$\mathbf{X}^{(l)} = \text{FFN}(\text{ARC}_{\text{FFN}}(\text{LN}(\mathbf{X}^{(l)\prime}))) + \mathbf{X}^{(l)\prime}. \tag{8}$$

Note that $\text{ARC}_{\text{MHA}}$ and $\text{ARC}_{\text{FFN}}$ are two independent ARC modules, meaning that the projection matrices of the two modules are not shared between them. During the fine-tuning phase, we exclusively update the learnable parameters of our newly added ARC modules. This involves freezing all original parameters of the pre-trained model and solely focusing on updating the parameters of our ARC.

**Inference.** Our ARC employs a completely linear transformation so that we can re-parameterize it by fusing the module to the original framework of the pre-trained model. Take the ARC module of FNN as an example, the process can be defined by:

$$\mathbf{X}^{(l)} = \text{GELU}(\text{ARC}_{\text{FFN}}(\mathbf{X}^{(l)\prime})\mathbf{W}_1^{(l)})\mathbf{W}_2^{(l)}, \tag{9}$$

where $\text{ARC}_{\text{FFN}}(\mathbf{X}^{(l)\prime})$ can be represented by $\mathbf{X}^{(l)\prime}(\mathbf{W}_{\text{down}}\mathbf{C}^{(l)}\mathbf{W}_{\text{up}} + \mathbf{I})$ according to Eq. (7) with $\mathbf{I} \in \mathbb{R}^{D \times D}$ being an identity matrix. Furthermore, we can construct $\mathbf{W}_1^{(l)\prime}$ by:

$$\mathbf{W}_1^{(l)\prime} = (\mathbf{W}_{\text{down}}\mathbf{C}^{(l)}\mathbf{W}_{\text{up}} + \mathbf{I})\mathbf{W}_1^{(l)}. \tag{10}$$

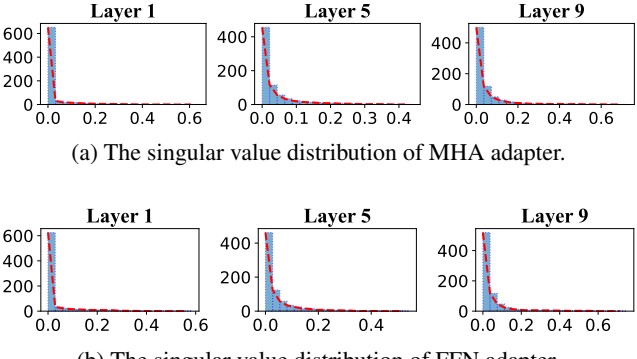

(a) The singular value distribution of MHA adapter.

(b) The singular value distribution of FFN adapter.

Figure 3: Singular value distribution of adaptation matrices without the bottleneck structure. Two adaptation matrices of both MHA and FFN blocks are fine-tuned on the *DTD* downstream task. The X-axis represents the singular values, while the Y-axis represents the count of singular values within specific ranges. Complete visualization is available in the appendix.

Therefore, we can replace the matrix $\mathbf{W}_1^{(l)}$ by $\mathbf{W}_1^{(l)\prime}$ in the inference stage, thereby avoiding extra computational overheads.

### 3.3 Insights of architecture design

In this work, we propose an approach that involves adopting a low-rank design for the adapters and sharing the bottleneck projections across layers. Our motivation for adopting this approach stems from the assumption that the layer-wise adapters can be effectively re-composed by re-scaling a small number of base linear projections. To validate this assumption, we conduct an analysis of the singular value distribution of adaptation matrices $\mathbf{W}_{\text{full}} \in \mathbb{R}^{D \times D}$ learned without the bottleneck operation, which are anticipated to be full-rank. In Fig. 3, we observe that the singular values exhibit extreme sparsity and follow a power-law distribution, with the majority of singular values being close to zero. This indicates that the learned adaptation matrices naturally exhibit low-rank characteristics. Furthermore, the pronounced sparsity of the singular values suggests that the adaptation matrices can be effectively reconstructed using a limited number of basis vectors. These findings provide strong evidence supporting the rationale behind our adaptation parameter-sharing strategy.

## 4 Experiments

In this section, we present the experimental settings, comparison to existing solutions, and ablation studies to unveil the key properties of the proposed method.

### 4.1 Experimental settings

**Datasets.** We evaluate the effectiveness of our ARC approach on two sets of visual task adaptation benchmarks, comprising a total of 24 datasets. The list of datasets used for evaluation is provided below:

*FGVC.* We conduct experiments with the default settings in the VPT [6] on a collection of five Fine-Grained Visual Classification (FGVC) datasets, known as FGVC. The FGVC dataset collection includes CUB-200-2011 [31], NABirds [32], Oxford Flowers [33], Stanford Dogs [34], and Stanford Cars [35].

*VTAB-1k.* We also evaluate our ARC method on the VTAB-1k benchmark [36], which consists of 19 diverse visual classification tasks. These tasks are divided into three groups: the *Natural* group, which contains images captured through standard cameras; the *Specialized* group, which includes images captured by specialist equipment such as remote sensing and medical imaging; and the *Structured* group, which comprises synthesized images from simulated environments, such as object counting and 3D depth prediction. Each downstream task in the VTAB-1k benchmark consists of 1000 training examples. Following VPT [6], we set aside 200 samples from the training set as the validation set to

Table 1: Comparison of ARC with baselines and state-of-the-art efficient adaptation methods on five FGVC datasets. All methods utilize ViT-B/16 pre-trained on ImageNet-21k as the backbone. "SSF*" denotes the performance reported in the original SSF paper [9], which incorporates advanced data augmentations like cutmix [39], mixup [40], and regularization techniques such as label smoothing [41]. To ensure a fair comparison, we reproduced the SSF method using the code provided by [9], while employing the same basic data augmentations as our approach, and we denote SSF's reported performance as "SSF*" and ARC's performance augmented with SSF's data augmentation as "ARC*". The **bold** font shows the best accuracy of all methods and the underline font shows the second best accuracy.

| Method \ Dataset | CUB-200-2011 | NABirds | Oxford Flowers | Stanford Dogs | Stanford Cars | Mean | Params.(M) |
|---|---|---|---|---|---|---|---|
| Full fine-tuning | 87.3 | 82.7 | 98.8 | 89.4 | 84.5 | 88.5 | 85.98 |
| Linear probing | 85.3 | 75.9 | 97.9 | 86.2 | 51.3 | 79.3 | 0.18 |
| Adapter [7] | 87.1 | 84.3 | 98.5 | 89.8 | 68.6 | 85.7 | 0.41 |
| Bias [42] | 88.4 | 84.2 | 98.8 | 91.2 | 79.4 | 88.4 | 0.28 |
| VPT-Shallow [6] | 86.7 | 78.8 | 98.4 | 90.7 | 68.7 | 84.6 | 0.25 |
| VPT-Deep [6] | **88.5** | 84.2 | 99.0 | 90.2 | 83.6 | 89.1 | 0.85 |
| LoRA [24] | 88.3 | 85.6 | 99.2 | 91.0 | 83.2 | 89.5 | 0.44 |
| SSF [9] | 82.7 | **85.9** | 98.5 | 87.7 | 82.6 | 87.5 | 0.39 |
| SSF* [9] | 89.5 | 85.7 | 99.6 | 89.6 | 89.2 | 90.7 | 0.39 |
| ARC$_{att}$ | 88.4 | 85.0 | **99.4** | 90.1 | 82.7 | 89.1 | 0.22 |
| ARC | **88.5** | 85.3 | 99.3 | **91.9** | **85.7** | **90.1** | 0.25 |
| ARC* | 89.3 | 85.7 | 99.7 | 89.1 | 89.5 | 90.7 | 0.25 |

select hyperparameters. Subsequently, we train the model on the full training data using the selected hyperparameters.

**Pre-trained backbone.** To evaluate the adaptation capacity of the proposed ARC method, we apply the ARC strategy to two typical types of Vision Transformers: ViT [1] and Swin Transformers [11]. For ViT, we conduct experiments using three different backbone variants with varying model sizes: ViT-**B**ase/**L**arge/**H**uge. All the backbones are pre-trained on the ImageNet-21K [15] dataset.

**Baselines and existing methods.** In our comparative analysis, we evaluate the performance of the ARC method against two baselines and several state-of-the-art efficient pre-trained model adaptation methods. The two baselines we consider are: (1) Full Fine-tuning: This baseline involves updating all the parameters of the pre-trained model using the training data of the downstream task. (2) Linear Probing: This baseline focuses on learning a linear classification head on the downstream task while keeping the remaining parameters of the pre-trained model frozen. In addition to the baselines, we compare our ARC method with the following state-of-the-art solutions: (1) Adapter [7]: This method inserts lightweight adaptation operations, consisting of a down-projection, non-linear activation, and an up-projection, into the pre-trained model. (2) Bias [37]: The Bias method fine-tunes only the bias terms of the pre-trained models while keeping the remaining parameters frozen. (3) LoRA [24]: This approach introduces trainable low-rank adaptation matrices into each layer of the Transformer architecture. (4) VPT [6]: The VPT method incorporates extra learnable tokens into the input or all attention layers of the frozen Transformer. (5) SSF [9]: This method adds linear transformation parameters, including scaling and shifting, to modulate the pre-trained features. By comparing the performance of our ARC method with these baselines and state-of-the-art solutions, we aim to demonstrate its superiority in terms of efficiency and effectiveness in pre-trained model adaptation.

**Implementation details.** To ensure a fair evaluation of the effectiveness of our proposed ARC method, we have opted for a simple training setup without too many additional bells and whistles. Similar to VPT [6], we have used standard data augmentations during the training phase, which include image normalization using ImageNet means and standard deviation, random resize crop to $224 \times 224$ with random horizontal flip for FGVC datasets, and resize to $224 \times 224$ for VTAB-1k. We have used grid search to select hyper-parameters such as the learning rate, weight decay, and batch size, using the validation set of each task, as in VPT [6]. All experiments were conducted using the PyTorch [38] framework on an NVIDIA A40 GPU with 48GB of GPU memory. Further details can be found in the appendix.

Table 2: Comparison of ARC with baselines and state-of-the-art efficient adaptation methods on VTAB-1k benchmark. All methods utilize ViT-B/16 pre-trained on ImageNet-21k as the backbone. To ensure a fair comparison, we reproduced the SSF method using the code provided by [9], while employing the same basic data augmentations as our approach.

| Method \ Dataset | Natural | | | | | | | | Specialized | | | | | Structured | | | | | | | | | Mean Total | Params.(M) |
|---|---|---|---|---|---|---|---|---|---|---|---|---|---|---|---|---|---|---|---|---|---|---|---|---|
| | CIFAR-100 | Caltech101 | DTD | Flowers102 | Pets | SVHN | Sun397 | Mean | Camelyon | EuroSAT | Resisc45 | Retinopathy | Mean | Clevr-Count | Clevr-Dist | DMLab | KITTI-Dist | dSpr-Loc | dSpr-Ori | sNORB-Azim | sNORB-Ele | Mean | | |
| Full fine-tuning | 68.9 | 87.7 | 64.3 | 97.2 | 86.9 | 87.4 | 38.8 | 75.9 | 79.7 | 95.7 | 84.2 | 73.9 | 83.4 | 56.3 | 58.6 | 41.7 | 65.5 | 57.5 | 46.7 | 25.7 | 29.1 | 47.6 | 65.6 | 85.8 |
| Linear probing | 63.4 | 85.0 | 63.2 | 97.0 | 86.3 | 36.6 | 51.0 | 68.9 | 78.5 | 87.5 | 68.6 | 74.0 | 77.2 | 34.3 | 30.6 | 33.2 | 55.4 | 12.5 | 20.0 | 9.6 | 19.2 | 26.9 | 52.9 | 0.04 |
| Adapter [7] | 74.1 | 86.1 | 63.2 | 97.7 | 87.0 | 34.6 | 50.8 | 70.5 | 76.3 | 88.0 | 73.1 | 70.5 | 77.0 | 45.7 | 37.4 | 31.2 | 53.2 | 30.3 | 25.4 | 13.8 | 22.1 | 32.4 | 55.8 | 0.27 |
| Bias [42] | 72.8 | 87.0 | 59.2 | 97.5 | 85.3 | 59.9 | 51.4 | 73.3 | 78.7 | 91.6 | 72.9 | 69.8 | 78.3 | 61.5 | 55.6 | 32.4 | 55.9 | 66.6 | 40.0 | 15.7 | 25.1 | 44.1 | 62.1 | 0.14 |
| VPT-Shallow [6] | 77.7 | 86.9 | 62.6 | 97.5 | 87.3 | 74.5 | 51.2 | 76.8 | 78.2 | 92.0 | 75.6 | 72.9 | 79.7 | 50.5 | 58.6 | 40.5 | 67.1 | 68.7 | 36.1 | 20.2 | 34.1 | 47.0 | 64.9 | 0.11 |
| VPT-Deep [6] | 78.8 | 90.8 | 65.8 | 98.0 | 88.3 | 78.1 | 49.6 | 78.5 | 81.8 | 96.1 | 83.4 | 68.4 | 82.4 | 68.5 | 60.0 | 46.5 | 72.8 | 73.6 | 47.9 | 32.9 | 37.8 | 55.0 | 69.4 | 0.60 |
| LoRA [24] | 65.3 | 87.9 | 69.4 | 98.7 | 90.7 | 82.4 | 53.4 | 78.2 | 82.8 | 94.8 | 82.5 | 75.0 | 83.8 | 77.6 | 64.7 | 45.8 | 79.0 | 73.3 | 44.7 | 26.3 | 38.2 | 56.2 | 70.1 | 0.29 |
| SSF [9] | 58.0 | 89.8 | 70.5 | 98.9 | 90.2 | 90.5 | 52.9 | 78.7 | 86.7 | 95.2 | 86.4 | 75.4 | 85.9 | 68.2 | 61.0 | 52.8 | 80.7 | 77.3 | 48.5 | 27.6 | 31.1 | 55.9 | 70.6 | 0.24 |
| SSF* [9] | 69.0 | 92.6 | 75.1 | 99.4 | 91.8 | 90.2 | 52.9 | 81.6 | 87.4 | 95.9 | 87.4 | 75.5 | 86.6 | 75.9 | 62.3 | 53.3 | 80.6 | 77.3 | 54.9 | 29.5 | 37.9 | 59.0 | 73.1 | 0.24 |
| ARC$_{att}$ | 70.1 | 90.5 | 70.5 | 98.8 | 90.8 | 88.6 | 53.6 | 80.4 | 84.6 | 95.5 | 86.6 | 75.5 | 85.6 | 79.0 | 65.6 | 48.6 | 81.3 | 75.1 | 48.7 | 29.1 | 39.6 | 58.4 | 72.2 | 0.08 |
| ARC | 72.2 | 90.1 | 72.7 | 99.0 | 91.0 | 91.9 | 54.4 | 81.6 | 84.9 | 95.7 | 86.7 | 75.8 | 85.8 | 80.7 | 67.1 | 48.7 | 81.6 | 79.2 | 51.0 | 31.4 | 39.9 | 60.0 | 73.4 | 0.13 |
| ARC* | 71.2 | 90.9 | 75.9 | 99.5 | 92.1 | 90.8 | 52.0 | 81.8 | 87.4 | 96.5 | 87.6 | 76.4 | 87.0 | 83.3 | 61.1 | 54.6 | 81.7 | 81.0 | 57.0 | 30.9 | 41.3 | 61.4 | 74.3 | 0.13 |

## 4.2 Experimental comparisons

In this section, we provide a comprehensive comparison of our ARC method with baseline methods and other state-of-the-art solutions. We evaluate the performance in terms of classification accuracy on downstream tasks as well as the parameter size. The results are summarized in Table 1 and Table 2, with all results obtained using the ViT-B/16 backbone. Based on these comparative results, we make the following observations:

(1) The ARC method demonstrates highly competitive classification accuracy on both sets of visual adaptation datasets, while maintaining a low parameter size. As shown in Table 1 and Table 2, under a fair comparison, ARC achieves the best mean accuracy and outperforms the other methods on the majority of the 24 datasets. This confirms the effectiveness of our proposed pre-trained model adaptation strategy. Furthermore, thanks to the parameter-sharing strategy in ARC, we are able to noticeably reduce the parameter size compared to other rank-decomposition based adapters such as Adapter [7] and LoRA [24]. VPT-Shallow [6] also exhibits parameter efficiency as it only introduces learnable tokens in the input layer. However, this is achieved at the cost of a significant performance sacrifice, resulting in considerably inferior performance compared to VPT-deep [6] and ARC. Another parameter-efficient method, Bias [42], focuses on updating only the bias terms in the pre-trained network, but it also leads to a significant compromise in classification performance on downstream tasks. To further decrease the parameter size, we evaluate ARC$_{att}$, which omits the adapters applied to Feed-Forward Network (FFN) blocks and focuses solely on adapting the Multi-Head Attention (MHA) blocks. This approach achieves nearly a halving of the parameter size while experiencing only a minimal $1\%$ drop in performance.

(2) In comparison to the efficient adaptation solutions presented in the tables, full fine-tuning yields comparable or even worse classification accuracy across various downstream tasks, despite updating a significantly larger number of parameters. This observation further emphasizes the importance and potential of lightweight adaptation designs. On the other end of the spectrum, linear probing requires minimal parameters but exhibits noticeable performance degradation.

**Experiments on larger-scale ViT backbones.** In addition to evaluating the ARC method on ViT-B/16 backbone, we also conducted experiments on larger-scale ViT backbones to assess its performance on more computationally demanding models. Specifically, we tested the ARC method on ViT-Large and ViT-Huge backbones. The results, presented in Table 3a and Table 3b, demonstrate that the ARC method maintains its competitive classification accuracy even with larger-scale backbones. It consistently outperforms the baseline methods and achieves comparable or superior performance compared to other state-of-the-art adaptation methods. Furthermore, the parameter size of the ARC method remains noticeably smaller than rank-decomposition based adapters like Adapter [7] and LoRA [24], as well as VPT-deep [6], showcasing its efficiency in terms of parameter utilization. These findings suggest that the ARC method is not only effective on smaller-scale ViT backbones

Table 3: Performance comparison on VTAB-1k using ViT-Large and ViT-Huge pre-trained on ImageNet-21k as backbone. "(·)" denotes the number of tasks in the subgroup. Expanded results are presented in the appendix.

(a) ViT-Large

| | Natural (7) | Specialized (4) | Structed (8) | Mean Total | Params. |
|---|---|---|---|---|---|
| Full fine-tuning | 74.7 | 83.8 | 48.1 | 65.4 | 303.4 |
| Linear probing | 70.9 | 69.1 | 25.8 | 51.5 | 0.05 |
| Adapter [7] | 68.6 | 73.5 | 29.0 | 52.9 | 2.38 |
| Bias [37] | 70.5 | 73.8 | 41.2 | 58.9 | 0.32 |
| VPT-Shallow [6] | 78.7 | 79.9 | 40.6 | 62.9 | 0.15 |
| VPT-Deep [6] | **82.5** | 83.9 | 54.1 | 70.8 | 0.49 |
| LoRA [24] | 81.4 | **85.0** | **57.3** | 72.0 | 0.74 |
| ARC | 82.3 | **85.6** | **57.3** | **72.5** | 0.18 |

(b) ViT-Huge

| | Natural (7) | Specialized (4) | Structed (8) | Mean Total | Params. |
|---|---|---|---|---|---|
| Full fine-tuning | 70.9 | 83.6 | 46.0 | 63.1 | 630.9 |
| Linear probing | 67.9 | 79.0 | 26.1 | 52.7 | 0.06 |
| Adapter [7] | 68.1 | 76.4 | 24.5 | 51.5 | 5.78 |
| Bias [42] | 70.3 | 78.9 | 41.7 | 60.1 | 0.52 |
| VPT-Shallow [6] | 74.8 | 81.2 | 43.0 | 62.8 | 0.18 |
| VPT-Deep [6] | 77.9 | 83.3 | 52.2 | 68.2 | 0.96 |
| LoRA [24] | 77.1 | 83.5 | **55.4** | 69.3 | 1.21 |
| ARC | **79.1** | 84.8 | 53.7 | 69.6 | 0.22 |

Table 4: Performance comparison on VTAB-1k using Swin-Base pre-trained on ImageNet-21k as backbone. "(·)" denotes the number of tasks in the subgroup. Expanded results are presented in the appendix.

| | Natural (7) | Specialized (4) | Structed (8) | Mean Total | Params. |
|---|---|---|---|---|---|
| Full fine-tuning | 79.1 | 86.2 | 59.7 | 72.4 | 86.8 |
| Linear probing | 73.5 | 80.8 | 33.5 | 58.2 | 0.05 |
| MLP-4 [6] | 70.6 | 80.7 | 31.2 | 57.7 | 4.04 |
| Partial [6] | 73.1 | 81.7 | 35.0 | 58.9 | 12.65 |
| Bias [42] | 74.2 | 80.1 | 42.4 | 62.1 | 0.25 |
| VPT-Shallow [6] | **79.9** | 82.5 | 37.8 | 62.9 | 0.05 |
| VPT-Deep [6] | 76.8 | 84.5 | 53.4 | 67.7 | 0.22 |
| ARC | 79.0 | **86.6** | **59.9** | **72.6** | 0.27 |

but also scalable to larger models, making it a promising solution for efficient pre-trained model adaptation across a wide range of backbone sizes.

**Experiments on hierarchical Vision Transformers.** We extended the ARC method to Swin Transformer [11], a hierarchical Transformer architecture. To accommodate the varying feature dimensionalities in Swin Transformer, we introduced a stage-sharing strategy, enabling parameter sharing within each stage. The results on the VTAB-1k benchmark (Table 4) demonstrate the generalizability of ARC. It achieves competitive transfer learning accuracy and maintains favorable parameter scale, except for the Natural group where ARC performs relatively weaker. These findings highlight ARC's versatility and effectiveness in adapting different transformer architectures, showcasing its potential for practical applications in visual adaptation.

## 4.3 Ablation studies

In order to gain deeper insights into the ARC method, we performed ablation studies to explore its additional properties. The experiments were conducted on the VTAB-1k benchmark using the pre-trained ViT-B/16 model. The results of these ablation studies are presented in Table 5.

**Bottleneck dimensionality.** In our ARC method, we adopt the low-rank design for the adapters but with the added feature of sharing the down-/up-projection matrices across layers and learning low-dimensional re-scaling coefficients to re-compose adaptation matrices. In this section, we investigate the impact of the bottleneck dimensionality on adaptation performance. The results are presented in Table 5a. We find that a dimensionality of 50 achieves the best balance between transfer learning performance and parameter efficiency. Further reduction to a 10-dimensional space leads to fewer parameters but at the cost of performance degradation. Conversely, higher-dimensional hidden spaces result in inferior performance. These findings validate the effectiveness of our low-rank design, with 50 linear projections providing sufficient flexibility for composing layer-adaptive adapters.

**Adapter positioning.** By default, our adapters are positioned before the MHA and FFN modules, allowing the adaptation operations to be seamlessly integrated into the pre-trained network during inference without additional inference cost. In this section, we investigate the impact of adapter

Table 5: Ablation experiments on VTAB-1k benchmark using ViT-B/16 backbone. The table shows average accuracy ("Acc.") and parameter count ("Params.") for all downstream datasets.

(a) Bottleneck dimension.

| Bott. dim | Acc. | Params. |
|---|---|---|
| 10 | 72.4 | 0.07 |
| 50 | **73.4** | 0.13 |
| 100 | 73.1 | 0.21 |
| 200 | 71.4 | 0.36 |

(b) ARC location.

| Location | Acc. | Params. |
|---|---|---|
| Before MHA | 72.2 | 0.08 |
| After MHA | 69.1 | 0.08 |
| Before FFN | 71.1 | 0.08 |
| After FFN | 69.0 | 0.08 |
| Before MHA & FFN | **73.4** | 0.13 |
| After MHA & FFN | 71.4 | 0.13 |

(c) Para. sharing strategy.

| Strategy | Acc. | Params. |
|---|---|---|
| non-intra + non-inter | **73.4** | 0.98 |
| intra + inter* | 72.9 | 0.10 |
| intra + inter | **73.4** | 0.13 |
| non-intra + inter | **73.4** | 0.21 |

(d) Insert selection.

| Layer ind. | Form | Acc. | Params. |
|---|---|---|---|
| $1 \sim 6$ | sequential | 71.5 | 0.126 |
| $7 \sim 12$ | sequential | 67.9 | 0.126 |
| $1 \sim 12$ | sequential | **73.4** | 0.133 |
| $1 \sim 12$ | parallel | 70.4 | 0.133 |

position on pre-trained model adaptation performance using different positioning strategies. The results are presented in Table 5b. Interestingly, placing the adapters after the MHA and/or FFN modules leads to performance degradation, despite this strategy being commonly adopted in previous works such as[7; 29]. Moreover, using only one type of adapter for either MHA or FFN results in inferior performance compared to using both types of adapters. This suggests that employing both types of adapters allows for more comprehensive adaptation of the pre-trained model to the target task without significantly increasing the parameter count.

**Sharing v.s non-sharing adaptation parameters.** In ARC, we adopt a parameter-sharing strategy to effectively reduce the number of adaptation parameters. This strategy encompasses two aspects: intra-layer sharing and inter-layer sharing. Through symmetric down-projection and up-projection matrices, we achieve intra-layer sharing, while inter-layer sharing involves sharing projection matrices across different layers. In this section, we investigate the impact of adaptation parameter sharing by conducting experiments with non-sharing or partial sharing designs. "intra + inter*" denotes sharing the bottleneck structure between MHA and FFN. The results presented in Table 5c demonstrate that using non-symmetric projection matrices or layer-independent adaptation parameters does not result in performance gains but leads to a noticeable increase in parameters. This validates the effectiveness of our parameter-sharing design.

**Adapter insertion.** We examine the performance of inserting the proposed adapters in a subset of layers using sequential or parallel insertion strategies. The results in Table 5d show that the performance of ARC improves as more layers are inserted. Furthermore, we observe that inserting adapters in the first six layers yields better results compared to inserting them in the last six layers. Additionally, we explore a parallel insertion setting inspired by [24], but the impact is not significantly pronounced. Another notable aspect is that our parameter sharing scheme in the ARC method prevents a linear increase in parameter size with the number of layers, ensuring better scalability, particularly for larger-scale models.

## 5 Limitation

The adaptation parameter sharing scheme in the ARC method is built on the assumption that layers have the same dimensionality. This assumption is crucial as it enables the sharing of down-/up-projection matrices involved in the bottleneck operation across layers, leading to parameter efficiency. However, it is worth exploring strategies to extend this scheme and accommodate dimensionality variation. This research direction holds promise for addressing scenarios where dimensionality varies and further enhancing the flexibility and applicability of the ARC method.

## 6 Conclusions

Our paper introduced the Adapter Re-Composing (ARC) method, which leverages the reusability of adaptation parameters to efficiently adapt pre-trained models. By sharing down-/up-projections in low-rank adapters across layers and learning layer-specific re-scaling coefficients to re-composing layer-adaptive adapters, ARC balances transfer learning performance and adaptation overheads. Extensive experiments demonstrate the compelling performance of our approach with a reduced parameter size. ARC offers a promising solution for efficient pre-trained model adaptation, showcasing the potential of reusing adaptation parameters for competitive results.

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

# Efficient Adaptation of Large Vision Transformer via Adapter Re-Composing

## Supplementary Materials

In the supplementary materials involving our work, we demonstrate detailed dataset settings, supplemental insights and analysis, extra experimental details, supplemental experiments, and broader impacts, including:

- A **Detailed descriptions for datasets and implementation**

- B **Insights of architecture design**

- C **Parameter size analysis**

- D **Experimental details on larger-scale and hierarchical ViT backbones**

- E **Experimental details on ablation studies**

- F **Expanded experiments with self-supervised pre-training**

- G **Broader impacts**

Due to the limitation that the file "Supplementary Materials.zip" larger than 100MB cannot be uploaded on OpenReview, the supplementary materials only upload the code for the project. Please refer to the anonymous link `https://drive.google.com/file/d/1ZblHbYF1JrOuOGeTLI4uII6GHt3CV3I2/view` to obtain the complete code, datasets, and models.

# A Detailed descriptions for datasets and implementation

We describe the details of visual adaptation classification tasks in Table 6 (FGVC) and 7 (VTAB-1k), including the class number and the train/val/test sets.

Table 6: Dataset statistics for FGVC. "*" denotes the train/val split of datasets following the dataset setting of VPT models [6].

| Dataset | Description | Classes | Train size | Val size | Test size |
|---|---|---|---|---|---|
| CUB-200-2011 [31] | Fine-grained bird species recognition | 200 | 5,394* | 600* | 5,794 |
| NABirds [32] | Fine-grained bird species recognition | 555 | 21,536* | 2,393* | 24,633 |
| Oxford Flowers [33] | Fine-grained flower species recognition | 102 | 1,020 | 1,020 | 6,149 |
| Stanford Dogs [34] | Fine-grained dog species recognition | 120 | 10,800* | 1,200* | 8,580 |
| Stanford Cars [35] | Fine-grained car classificatio | 196 | 7,329* | 815* | 8,041 |

Table 7: Dataset statistics for VTAB-1k [36].

| Dataset | Description | Classes | Train size | Val size | Test size |
|---|---|---|---|---|---|
| CIFAR-100 | | 100 | | | 10,000 |
| Caltech101 | | 102 | | | 6,084 |
| DTD | | 47 | | | 1,880 |
| Flowers102 | Natural | 102 | 800/1,000 | 200 | 6,149 |
| Pets | | 37 | | | 3,669 |
| SVHN | | 10 | | | 26,032 |
| Sun397 | | 397 | | | 21,750 |
| Patch Camelyon | | 2 | | | 32,768 |
| EuroSAT | | 10 | | | 5,400 |
| Resisc45 | Specialized | 45 | 800/1,000 | 200 | 6,300 |
| Retinopathy | | 5 | | | 42,670 |
| Clevr/count | | 8 | | | 15,000 |
| Clevr/distance | | 6 | | | 15,000 |
| DMLab | | 6 | | | 22,735 |
| KITTI/distance | | 4 | | | 711 |
| dSprites/location | Structured | 16 | 800/1,000 | 200 | 73,728 |
| dSprites/orientation | | 16 | | | 73,728 |
| SmallNORB/azimuth | | 18 | | | 12,150 |
| SmallNORB/elevation | | 9 | | | 12,150 |

Table 8 summarizes the detailed configurations we used for experiments. As mentioned in Section 4.1, we utilize grid search to select hyper-parameters such as learning rate, weight decay, batch size, and adapter dropout, using the validation set of each task. Note that we also apply dropout to the middle features produced by our ARC method, which we term as "adapter dropout". Specifically, during the ARC process, we randomly drop partial features before up-projection.

# B Insights of architecture design

Similar to Fig. 3, we present more visualization results of singular value distribution of adaptation matrices $\mathbf{W}_{\text{full}} \in \mathbb{R}^{D \times D}$ learned without the bottleneck operation. As shown in Fig. 4, the singular value distribution of adaptation matrices learned on *DTD* downstream task exhibits a power-law

Table 8: The implementation details of configurations such as optimizer and hyper-parameters. We select the best hyper-parameters for each download task via using grid search.

| Optimizer | AdamW |
|---|---|
| Learning Rate | {0.2, 0.1, 0.05, 0.01, 0.005, 0.001, 0.0001} |
| Weight Decay | {0.05, 0.01, 0.005, 0.001, 0} |
| Batch Size | {256, 128, 32} |
| Adapter Dropout | {0.8, 0.5, 0.1, 0} |
| Learning Rate Schedule | Cosine Decay |
| Training Epochs | 100 |
| Warmup Epochs | 10 |

distribution across various layers in the downstream tasks. This finding provides further support for our research motivation.

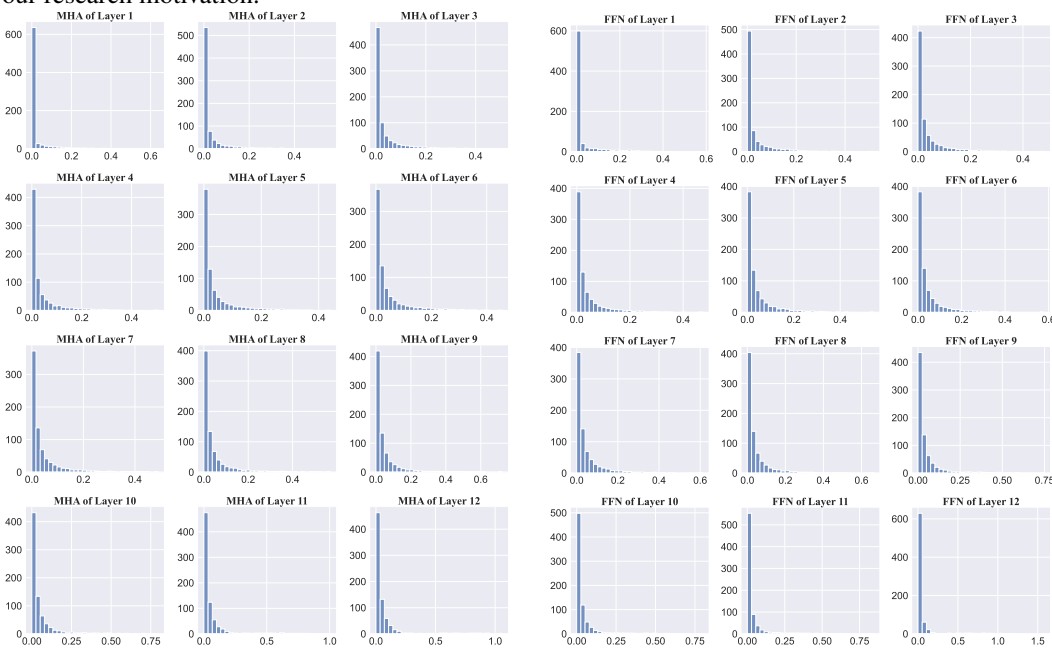

(a) Singular value distribution of MHA adapter.   (b) Singular value distribution of FFN adapter.

Figure 4: Singular value distribution of adaptation matrices without the bottleneck structure. Two adaptation matrices of both MHA and FFN blocks are fine-tuned on the *DTD* downstream task. The X-axis represents the singular values, while the Y-axis represents the count of singular values within specific ranges.

## C  Parameter size analysis

To showcase the parameter-efficiency of our ARC method, we compare its parameter size with other popular lightweight adaptation methods (Table 9), including Adapter [7], VPT [6], LoRA [24], and SSF [9]. Adapter [7] adds two linear projections to each encoder layer during fine-tuning, resulting in the introduction of $2 \cdot D \cdot D' \cdot L$ learnable parameters, where $D'$ denotes the hidden dimensionality of the linear projections. Furthermore, due to the presence of non-linear activations in Adapter, the additional parameters contribute to supernumerary overhead during the inference phase. VPT [6] incorporates $m$ prompts into input space, leading to an increase of $m \cdot D$ parameters for VPT-Shallow and $m \cdot D \cdot L$ parameters for VPT-Deep. In contrast to Adapter, both LoRA [24] and SSF [9] employ linear adaptation methods without incorporating non-linear functions. This design choice allows them to leverage re-parameterization benefits, thereby mitigating additional computations during inference. Specifically, the adaptation matrix of LoRA, which consists of a down-projection and an up-projection, introduces $2 \cdot w \cdot D \cdot D' \cdot L$ learnable parameters, where $w$ denotes the number of attention matrices undergoing adaptation. SSF inserts linear scaling and shifting coefficients after

$o$ operations, resulting in an addition of $2 \cdot o \cdot D \cdot L$ extra parameters. The proposed ARC method offers additional parameter compression by sharing symmetric projection matrices across different layers. This approach introduces only $D \cdot D'$ parameters. Additionally, we learn low-dimensional re-scaling coefficients and bias terms for each layer, resulting in a total of $(D' + D) \cdot L$ additional parameters. Overall, the number of parameters in our default ARC is $2 \cdot ((D \cdot D') + (D' + D) \cdot L)$.

Table 9: Comparison of the additional parameter size in both fine-tuning and inference stages with other lightweight adaptation methods.

| Method Stage | Adapter [7] | VPT-Shallow [6] | VPT-Deep [6] | LoRA [24] | SSF [9] | ARC |
|---|---|---|---|---|---|---|
| Fine-Tuning | $2 \cdot D \cdot D' \cdot L$ | $m \cdot D$ | $m \cdot D \cdot L$ | $2 \cdot w \cdot D \cdot D' \cdot L$ | $2 \cdot o \cdot D \cdot L$ | $2 \cdot (D \cdot D' + (D' + D) \cdot L)$ |
| Inference | $2 \cdot D \cdot D' \cdot L$ | $m \cdot D$ | $m \cdot D \cdot L$ | 0 | 0 | 0 |

We also compare the parameter size with lightweight adaptation methods on backbones of different scales, as shown in Fig. 5. Our ARCs demonstrate parameter efficiency across various model sizes, comparable to VPT-Shallow [6]. However, the unique advantage of our approach lies in its ability to effectively balance lower overheads and maintain competitive performance. Furthermore, the parameter count of our ARC remains stable even as the model scale increases, showcasing the scalability of our method with minimal additional resource consumption.

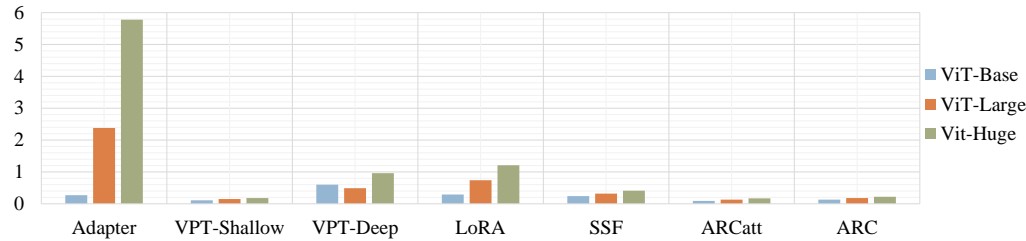

Figure 5: The parameter size comparison of lightweight adaptation methods on ViT Backbones of Different Scales. The X-axis represents different adaptation methods, while the Y-axis represents the parameter size in Million (M).

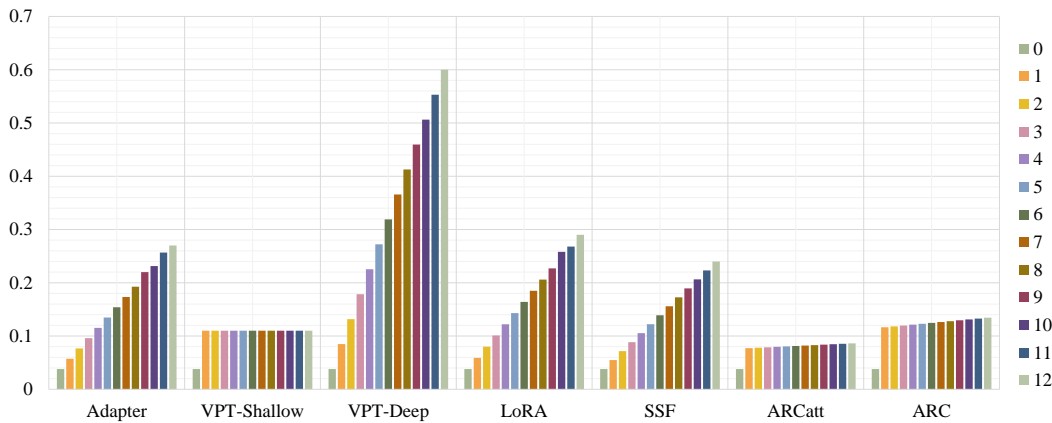

Figure 6: The parameter size comparison with lightweight adaptation methods with a different number of inserted layers. The X-axis represents different adaptation methods, while the Y-axis represents the parameter size in Million (M).

Thanks to our adaptation parameter sharing strategy, the ARC method avoids a linear increase in the number of learnable parameters as the number of layers grows. We employ ViT-B as the backbone and integrate adapters into different layers. As shown in Fig.6, in contrast to other adaptation methods, both our ARCs and VPT-Shallow[6] effectively manage parameter growth as the number of inserted layers increases, but only our methods achieve promising performance without significant cost escalation. This highlights the scalability and effectiveness advantages of our ARCs.

## D Experimental details on larger-scale and hierarchical ViT backbones

Table 10, 11 and 12 respectively display the comprehensive results of the comparison conducted in Section 4.2 among ViT-Large, ViT-Huge, and Swin-Base models.

Table 10: This table is extended from Table 3a in Section 4.2 and describes the detailed experimental results of the performance comparison on VTAB-1k using ViT-Large pre-trained on ImageNet-21k as the backbone.

| Method \ Dataset | Natural | | | | | | | | Specialized | | | | | Structured | | | | | | | | | Mean Total | Params.(M) |
|---|---|---|---|---|---|---|---|---|---|---|---|---|---|---|---|---|---|---|---|---|---|---|---|---|
| | CIFAR-100 | Caltech101 | DTD | Flowers102 | Pets | SVNH | Sun397 | Mean | Camelyon | EuroSAT | Resisc45 | Retinopathy | Mean | Clevr-Count | Clevr-Dist | DMLab | KITTI-Dist | dSpr-Loc | dSpr-Ori | sNORB-Azim | sNORB-Ele | Mean | | |
| Full fine-tuning | 68.6 | 84.3 | 58.6 | 96.3 | 86.5 | 87.5 | 41.4 | 74.7 | 82.6 | 95.9 | 82.4 | 74.2 | 83.8 | 55.4 | 55.0 | 42.2 | 74.2 | 56.8 | 43.0 | 28.5 | 29.7 | 48.1 | 65.4 | 303.4 |
| Linear probing | 72.2 | 86.4 | 63.6 | 97.4 | 85.8 | 38.1 | 52.5 | 70.9 | 76.9 | 87.3 | 66.6 | 45.4 | 69.1 | 28.2 | 28.0 | 34.7 | 54.0 | 10.6 | 14.2 | 14.6 | 21.9 | 25.8 | 51.5 | 0.05 |
| Adapter [7] | 75.3 | 84.2 | 54.5 | 97.4 | 84.3 | 31.3 | 52.9 | 68.6 | 75.8 | 85.1 | 63.4 | 69.5 | 73.5 | 35.4 | 34.1 | 30.8 | 47.1 | 30.4 | 23.4 | 10.8 | 19.8 | 29.0 | 52.9 | 2.38 |
| Bias [37] | 71.0 | 82.4 | 51.3 | 96.3 | 83.2 | 59.5 | 49.9 | 70.5 | 72.9 | 87.9 | 63.1 | 71.3 | 73.8 | 51.2 | 50.7 | 33.5 | 54.8 | 65.9 | 37.3 | 13.7 | 22.2 | 41.2 | 58.9 | 0.32 |
| VPT-Shallow [6] | 80.6 | 88.2 | 67.1 | 98.0 | 85.9 | 78.4 | 53.0 | 78.7 | 79.7 | 93.5 | 73.4 | 73.1 | 79.9 | 41.5 | 52.5 | 32.3 | 64.2 | 48.3 | 35.3 | 21.6 | 28.8 | 40.6 | 62.9 | 0.15 |
| VPT-Deep [6] | 84.1 | 88.9 | 70.8 | 98.8 | 90.0 | 89.0 | 55.9 | 82.5 | 82.5 | 96.6 | 82.6 | 73.9 | 83.9 | 63.7 | 60.7 | 46.1 | 75.7 | 83.7 | 47.4 | 18.9 | 36.9 | 54.1 | 70.8 | 0.49 |
| LoRA [24] | 75.8 | 89.8 | 73.6 | 99.1 | 90.8 | 83.2 | 57.5 | 81.4 | 86.0 | 95.0 | 83.4 | 75.5 | 85.0 | 78.1 | 60.5 | 46.7 | 81.6 | 76.7 | 51.3 | 28.0 | 35.4 | 57.3 | 72.0 | 0.74 |
| ARC_att | 75.6 | 89.9 | 72.2 | 99.0 | 90.4 | 89.0 | 57.5 | 81.9 | 86.1 | 95.0 | 85.4 | 76.0 | 85.6 | 75.0 | 60.1 | 48.0 | 80.9 | 77.0 | 51.3 | 27.2 | 35.6 | 56.9 | 72.2 | 0.13 |
| ARC | 76.2 | 89.6 | 73.4 | 99.1 | 90.3 | 90.9 | 56.5 | 82.3 | 85.0 | 95.7 | 85.9 | 75.8 | 85.6 | 78.6 | 62.1 | 46.7 | 76.7 | 75.9 | 53.0 | 30.2 | 35.2 | 57.3 | 72.5 | 0.18 |

Table 11: This table is extended from Table 3b in Section 4.2 and describes the detailed experimental results of the performance comparison on VTAB-1k using ViT-Huge pre-trained on ImageNet-21k as the backbone.

| Method \ Dataset | Natural | | | | | | | | Specialized | | | | | Structured | | | | | | | | | Mean Total | Params.(M) |
|---|---|---|---|---|---|---|---|---|---|---|---|---|---|---|---|---|---|---|---|---|---|---|---|---|
| | CIFAR-100 | Caltech101 | DTD | Flowers102 | Pets | SVNH | Sun397 | Mean | Camelyon | EuroSAT | Resisc45 | Retinopathy | Mean | Clevr-Count | Clevr-Dist | DMLab | KITTI-Dist | dSpr-Loc | dSpr-Ori | sNORB-Azim | sNORB-Ele | Mean | | |
| Full fine-tuning | 58.7 | 86.5 | 55.0 | 96.5 | 79.7 | 87.5 | 32.5 | 70.9 | 83.1 | 95.5 | 81.9 | 73.8 | 83.6 | 47.6 | 53.9 | 37.8 | 69.9 | 53.8 | 48.6 | 30.2 | 25.8 | 46.0 | 63.1 | 630.9 |
| Linear probing | 64.3 | 83.6 | 65.2 | 96.2 | 83.5 | 39.8 | 43.0 | 67.9 | 78.0 | 90.5 | 73.9 | 73.4 | 79.0 | 25.6 | 24.5 | 34.8 | 59.0 | 9.5 | 15.6 | 17.4 | 22.8 | 26.1 | 52.7 | 0.06 |
| Adapter [7] | 69.4 | 84.4 | 62.7 | 97.2 | 84.2 | 33.6 | 45.3 | 68.1 | 77.3 | 86.6 | 70.8 | 71.1 | 76.4 | 28.6 | 27.5 | 29.2 | 55.2 | 10.0 | 15.2 | 11.9 | 18.6 | 24.5 | 51.5 | 5.78 |
| Bias [37] | 65.7 | 84.3 | 59.9 | 96.6 | 80.6 | 60.1 | 44.9 | 70.3 | 79.7 | 92.8 | 71.5 | 71.6 | 78.9 | 52.3 | 50.4 | 31.2 | 57.7 | 65.9 | 39.7 | 16.7 | 20.2 | 41.7 | 60.1 | 0.52 |
| VPT-Shallow [6] | 70.6 | 84.7 | 64.8 | 96.4 | 85.1 | 75.6 | 46.2 | 74.8 | 79.9 | 93.7 | 77.7 | 73.6 | 81.2 | 40.3 | 60.9 | 34.9 | 63.3 | 61.3 | 38.9 | 19.8 | 24.9 | 43.0 | 62.8 | 0.18 |
| VPT-Deep [6] | 76.9 | 87.2 | 66.8 | 97.5 | 84.8 | 85.5 | 46.5 | 77.9 | 81.6 | 96.3 | 82.5 | 72.8 | 83.3 | 50.4 | 61.2 | 43.9 | 76.6 | 79.5 | 50.1 | 24.7 | 31.5 | 52.2 | 68.2 | 0.96 |
| LoRA [24] | 63.0 | 89.4 | 68.1 | 98.0 | 87.0 | 85.2 | 48.7 | 77.1 | 82.2 | 94.3 | 83.1 | 74.2 | 83.5 | 68.6 | 65.0 | 44.8 | 76.4 | 70.8 | 48.8 | 30.4 | 38.3 | 55.4 | 69.3 | 1.21 |
| ARC_att | 65.5 | 89.1 | 69.9 | 98.0 | 87.5 | 89.1 | 48.8 | 78.3 | 83.4 | 94.5 | 84.5 | 74.4 | 84.2 | 73.2 | 66.6 | 45.6 | 76.2 | 78.3 | 51.2 | 32.1 | 37.6 | 57.6 | 70.8 | 0.17 |
| ARC | 67.6 | 90.2 | 69.5 | 98.4 | 87.9 | 90.8 | 49.6 | 79.1 | 84.5 | 94.9 | 85.1 | 74.6 | 84.8 | 75.2 | 66.7 | 46.2 | 76.4 | 44.2 | 51.1 | 32.2 | 37.7 | 53.7 | 69.6 | 0.22 |

Table 12: This table is extended from Table 4 in Section 4.2 and describes the detailed experimental results of the performance comparison on VTAB-1k using Swin-Base pre-trained on ImageNet-21k as the backbone.

| Method \ Dataset | Natural | | | | | | | | Specialized | | | | | Structured | | | | | | | | | Mean Total | Params.(M) |
|---|---|---|---|---|---|---|---|---|---|---|---|---|---|---|---|---|---|---|---|---|---|---|---|---|
| | CIFAR-100 | Caltech101 | DTD | Flowers102 | Pets | SVNH | Sun397 | Mean | Camelyon | EuroSAT | Resisc45 | Retinopathy | Mean | Clevr-Count | Clevr-Dist | DMLab | KITTI-Dist | dSpr-Loc | dSpr-Ori | sNORB-Azim | sNORB-Ele | Mean | | |
| Full fine-tuning | 72.2 | 88.0 | 71.4 | 98.3 | 89.5 | 89.4 | 45.1 | 79.1 | 86.6 | 96.9 | 87.7 | 73.6 | 86.2 | 75.7 | 59.8 | 54.6 | 78.6 | 79.4 | 53.6 | 34.6 | 40.9 | 59.7 | 72.4 | 86.9 |
| Linear probing | 61.4 | 90.2 | 74.8 | 95.5 | 90.2 | 46.9 | 55.8 | 73.5 | 81.5 | 90.1 | 82.1 | 69.4 | 80.8 | 39.1 | 35.9 | 40.1 | 65.0 | 20.3 | 26.0 | 14.3 | 27.6 | 33.5 | 58.2 | 0.05 |
| MLP-4 [6] | 54.9 | 87.4 | 71.4 | 99.5 | 89.1 | 39.7 | 52.5 | 70.6 | 80.5 | 90.9 | 76.8 | 74.4 | 80.7 | 60.9 | 38.8 | 40.2 | 66.5 | 9.4 | 21.1 | 14.5 | 28.8 | 31.2 | 57.7 | 4.04 |
| Partial [6] | 60.3 | 88.9 | 72.6 | 98.7 | 89.3 | 50.5 | 51.5 | 73.1 | 82.8 | 91.7 | 80.1 | 72.3 | 81.7 | 34.3 | 35.5 | 43.2 | 77.1 | 15.8 | 26.2 | 19.1 | 28.4 | 35.0 | 58.9 | 12.65 |
| Bias [37] | 73.1 | 86.8 | 65.7 | 97.7 | 87.5 | 56.4 | 52.3 | 74.2 | 80.4 | 91.6 | 76.1 | 72.5 | 80.1 | 47.3 | 48.5 | 34.7 | 66.3 | 57.6 | 36.2 | 17.2 | 31.6 | 42.4 | 62.1 | 0.25 |
| VPT-Shallow [6] | 78.0 | 91.3 | 77.2 | 99.4 | 90.4 | 68.4 | 54.3 | 79.9 | 80.1 | 93.9 | 83.0 | 72.7 | 82.5 | 40.8 | 43.9 | 34.1 | 63.2 | 28.4 | 44.5 | 21.5 | 26.3 | 37.8 | 62.9 | 0.05 |
| VPT-Deep [6] | 79.6 | 90.8 | 78.0 | 99.5 | 91.4 | 46.5 | 51.7 | 76.8 | 84.9 | 96.2 | 85.0 | 72.0 | 84.5 | 67.6 | 59.4 | 50.1 | 74.1 | 74.4 | 50.6 | 25.7 | 25.7 | 53.4 | 67.7 | 0.22 |
| ARC_att | 67.2 | 89.7 | 74.7 | 99.5 | 89.7 | 88.5 | 52.7 | 80.3 | 88.1 | 95.9 | 85.7 | 77.2 | 86.7 | 76.5 | 58.5 | 52.1 | 82.8 | 89.4 | 56.4 | 27.5 | 35.1 | 59.8 | 73.0 | 0.16 |
| ARC | 62.5 | 90.0 | 71.9 | 99.2 | 87.8 | 90.7 | 51.1 | 79.0 | 89.1 | 95.8 | 84.5 | 77.0 | 86.6 | 75.4 | 57.4 | 53.4 | 83.1 | 91.7 | 55.2 | 31.6 | 31.8 | 59.9 | 72.6 | 0.27 |

## E Experimental details on ablation studies

Table 13, 14, 15 and 16 display the complete results of the ablation studies in Section 4.3.

Table 13: This table is extended from Table 5a in Section 4.3 and describes the detailed experimental content of the performance comparison among different bottleneck dimensionality.

| Dataset / Dimension | Natural | | | | | | | | Specialized | | | | | Structured | | | | | | | | | Mean Total | Params.(M) |
|---|---|---|---|---|---|---|---|---|---|---|---|---|---|---|---|---|---|---|---|---|---|---|---|---|
| | CIFAR-100 | Caltech101 | DTD | Flowers102 | Pets | SVNH | Sun397 | Mean | Camelyon | EuroSAT | Resisc45 | Retinopathy | Mean | Clevr-Count | Clevr-Dist | DMLab | KITTI-Dist | dSpr-Loc | dSpr-Ori | sNORB-Azim | sNORB-Ele | Mean | | |
| 10 | **72.2** | 88.4 | 71.2 | 98.7 | **91.1** | 89.4 | 54.7 | 80.9 | 84.7 | 95.6 | 86.0 | **75.8** | 85.6 | 80.1 | 65.9 | 48.8 | 80.5 | 75.5 | 48.3 | 30.2 | 38.6 | 58.5 | 72.4 | 0.07 |
| 50 | **72.2** | 90.1 | 72.7 | 99.0 | 91.0 | 91.9 | 54.4 | 81.6 | 84.9 | 95.7 | 86.7 | 75.8 | 85.8 | 80.7 | 67.1 | 48.7 | 81.6 | 79.2 | 51.0 | 31.4 | 39.9 | 60.0 | 73.4 | 0.13 |
| 100 | 71.3 | 90.0 | 73.0 | 99.0 | 90.7 | 91.8 | 55.1 | 81.6 | 85.1 | 96.3 | 86.1 | 75.4 | 85.7 | 80.8 | 67.2 | 49.0 | 79.3 | 74.8 | 50.1 | 34.0 | 39.1 | 59.3 | 73.1 | 0.21 |
| 200 | 70.5 | 89.3 | 72.9 | 99.1 | 89.8 | 91.9 | 54.9 | 81.2 | 84.9 | 95.3 | 84.0 | 75.7 | 85.0 | 80.0 | 67.8 | 48.9 | 76.8 | 50.8 | 51.3 | 34.4 | 39.1 | 56.1 | 71.4 | 0.36 |

Table 14: This table is extended from Table 5b in Section 4.3 and describes the detailed experimental content of the performance comparison among different adapter positioning.

| Dataset / Location | Natural | | | | | | | | Specialized | | | | | Structured | | | | | | | | | Mean Total | Params.(M) |
|---|---|---|---|---|---|---|---|---|---|---|---|---|---|---|---|---|---|---|---|---|---|---|---|---|
| | CIFAR-100 | Caltech101 | DTD | Flowers102 | Pets | SVNH | Sun397 | Mean | Camelyon | EuroSAT | Resisc45 | Retinopathy | Mean | Clevr-Count | Clevr-Dist | DMLab | KITTI-Dist | dSpr-Loc | dSpr-Ori | sNORB-Azim | sNORB-Ele | Mean | | |
| Before MHA | 70.1 | **90.5** | 70.5 | 98.8 | 90.8 | 88.6 | 53.6 | 80.4 | 84.6 | 95.5 | 86.6 | 75.5 | 85.6 | 79.0 | 65.6 | 48.6 | 81.3 | 75.1 | 48.7 | 29.1 | 39.6 | 58.4 | 72.2 | 0.08 |
| After MHA | 67.0 | 88.9 | 69.8 | 98.8 | 90.8 | 82.2 | 52.3 | 78.5 | 84.1 | 94.6 | 85.1 | 75.4 | 84.8 | 77.4 | 60.1 | 44.3 | 77.1 | 61.2 | 45.7 | 23.0 | 35.6 | 53.0 | 69.1 | 0.08 |
| Before FFN | 70.8 | 89.4 | 71.0 | 99.0 | 89.9 | 86.9 | 53.9 | 80.1 | 85.5 | 94.7 | 84.9 | 75.6 | 85.2 | 77.3 | 63.6 | 46.5 | 77.5 | 70.3 | 48.4 | 27.6 | 37.3 | 56.0 | 71.1 | 0.08 |
| After FFN | 66.7 | 88.2 | 69.6 | 98.6 | 90.2 | 82.5 | 52.9 | 78.4 | 83.6 | 94.8 | 85.3 | 75.5 | 84.8 | 77.9 | 63.1 | 44.1 | 76.7 | 57.9 | 47.0 | 22.6 | 33.9 | 52.9 | 69.0 | 0.08 |
| Before MHA & FFN | 72.2 | 90.1 | 72.7 | 99.0 | 91.0 | 91.9 | 54.4 | 81.6 | 84.9 | 95.7 | 86.7 | 75.8 | 85.8 | 80.7 | 67.1 | 48.7 | 81.6 | 79.2 | 51.0 | 31.4 | 39.9 | 60.0 | 73.4 | 0.13 |
| After MHA & FFN | 70.5 | 89.9 | 71.3 | 99.0 | 91.4 | 86.9 | 53.5 | 80.4 | 84.7 | 94.9 | 86.4 | 76.0 | 85.5 | 80.3 | 62.8 | 46.8 | 80.9 | 66.9 | 49.6 | 28.4 | 36.4 | 56.5 | 71.4 | 0.13 |

Table 15: This table is extended from Table 5c in Section 4.3 and describes the detailed experimental content of the performance comparison among different parameter sharing strategy.

| Dataset / Strategy | Natural | | | | | | | | Specialized | | | | | Structured | | | | | | | | | Mean Total | Params.(M) |
|---|---|---|---|---|---|---|---|---|---|---|---|---|---|---|---|---|---|---|---|---|---|---|---|---|
| | CIFAR-100 | Caltech101 | DTD | Flowers102 | Pets | SVNH | Sun397 | Mean | Camelyon | EuroSAT | Resisc45 | Retinopathy | Mean | Clevr-Count | Clevr-Dist | DMLab | KITTI-Dist | dSpr-Loc | dSpr-Ori | sNORB-Azim | sNORB-Ele | Mean | | |
| non-intra + non-inter | 70.1 | **91.1** | 71.5 | **99.2** | 90.6 | 91.9 | 54.6 | 81.3 | 84.8 | 95.5 | 86.4 | 75.4 | 85.5 | 81.1 | 66.1 | 50.1 | 78.6 | 80.3 | 51.5 | 35.8 | 40.6 | 60.5 | 73.4 | 0.98 |
| intra + inter* | 72.9 | 89.8 | 72.1 | 98.8 | 91.0 | 90.7 | 54.6 | 81.4 | 85.8 | 95.5 | 86.3 | 75.6 | 85.8 | 80.3 | 66.5 | 48.8 | 79.6 | 77.0 | 50.7 | 30.9 | 39.0 | 59.1 | 72.9 | 0.10 |
| intra + inter | 72.2 | 90.1 | 72.7 | 99.0 | 91.0 | 91.9 | 54.4 | 81.6 | 84.9 | 95.7 | 86.7 | 75.8 | 85.8 | 80.7 | 67.1 | 48.7 | 81.6 | 79.2 | 51.0 | 31.4 | 39.9 | 60.0 | 73.4 | 0.13 |
| non-intra + inter | 72.9 | 89.5 | 72.9 | 98.8 | 90.6 | 90.2 | 55.8 | 81.5 | 86.2 | 95.5 | 86.2 | 75.9 | 86.0 | 81.1 | 67.1 | 48.3 | 81.0 | 78.5 | 50.6 | 31.5 | 41.9 | 60.0 | 73.4 | 0.21 |

Table 16: This table is extended from Table 5d in Section 4.3 and describes the detailed experimental content of the performance comparison among different adapter insertion.

| Dataset / Layer Form | Natural | | | | | | | | Specialized | | | | | Structured | | | | | | | | | Mean Total | Params.(M) |
|---|---|---|---|---|---|---|---|---|---|---|---|---|---|---|---|---|---|---|---|---|---|---|---|---|
| | CIFAR-100 | Caltech101 | DTD | Flowers102 | Pets | SVNH | Sun397 | Mean | Camelyon | EuroSAT | Resisc45 | Retinopathy | Mean | Clevr-Count | Clevr-Dist | DMLab | KITTI-Dist | dSpr-Loc | dSpr-Ori | sNORB-Azim | sNORB-Ele | Mean | | |
| 1 ∼ 6 & sequential | 69.0 | 88.1 | 70.2 | 98.4 | 90.0 | 89.8 | 52.3 | 79.7 | 84.1 | 94.5 | 85.4 | 75.5 | 84.9 | 80.2 | 67.3 | 46.4 | 78.8 | 74.4 | 48.1 | 29.1 | 37.6 | 57.7 | 71.5 | 0.126 |
| 7 ∼ 12 & sequential | 57.9 | 88.2 | 68.4 | 98.3 | 89.2 | 70.3 | 52.1 | 74.9 | 82.2 | 94.3 | 84.3 | 76.4 | 84.3 | 77.0 | 58.1 | 45.4 | 75.3 | 74.0 | 42.7 | 21.0 | 34.9 | 53.6 | 67.9 | 0.126 |
| 1 ∼ 12 & sequential | 72.2 | 90.1 | 72.7 | 99.0 | 91.0 | 91.9 | 54.4 | 81.6 | 84.9 | 95.7 | 86.7 | 75.8 | 85.8 | 80.7 | 67.1 | 48.7 | 81.6 | 79.2 | 51.0 | 31.4 | 39.9 | 60.0 | 73.4 | 0.133 |
| 1 ∼ 12 & parallel | 70.7 | 90.9 | 71.5 | 98.9 | 91.1 | 86.1 | 53.8 | 80.4 | 83.5 | 95.1 | 85.6 | 75.4 | 84.9 | 76.6 | 64.1 | 45.9 | 76.9 | 62.0 | 46.0 | 25.3 | 37.2 | 54.3 | 70.4 | 0.133 |

# F  Expanded experiments with self-supervised pre-training

In addition to the models pre-trained with supervised objectives in Section 4, we also conduct experiments with self-supervised pre-training approaches: MAE [2] and Moco V3 [23]. Specifically, We utilize MAE [2] and Moco V3 [23] self-supervised pre-trained ViT-B as the backbone and evaluate the performance of our ARC on VTAB-1k. The results of MAE and Moco V3 self-supervised models are presented in Table 17 and Table 18, respectively. We observe that our ARC still exhibits competitive performance on two self-supervised ViTs. In addition, our ARC method outperforms other adaptation methods: Adapter[7] and LoRA [24] on the majority of downstream tasks. Surprisingly, the $ARC_{att}$ with smaller learnable parameters even surpasses the ARC across different self-supervised pre-trained models. A possible explanation could be that $ARC_{att}$ contains fewer parameters, which allows it to effectively prevent overfitting.

Table 17: Performance comparison on VTAB-1k using MAE self-supervised pre-trained ViT-Base as backbone.

| Method \ Dataset | Natural | | | | | | | | Specialized | | | | | Structured | | | | | | | | | Mean Total | Params.(M) |
|---|---|---|---|---|---|---|---|---|---|---|---|---|---|---|---|---|---|---|---|---|---|---|---|---|
| | CIFAR-100 | Caltech101 | DTD | Flowers102 | Pets | SVHN | Sun397 | Mean | Camelyon | EuroSAT | Resisc45 | Retinopathy | Mean | Clevr-Count | Clevr-Dist | DMLab | KITTI-Dist | dSpr-Loc | dSpr-Ori | sNORB-Azim | sNORB-Ele | Mean | | |
| Full fine tuning | 24.6 | 84.2 | 56.9 | 72.7 | 74.4 | 86.6 | 15.8 | 59.3 | 81.8 | 94.0 | 72.3 | 70.6 | 79.7 | 67.0 | 59.8 | 45.2 | 75.3 | 72.5 | 47.5 | 30.2 | 33.0 | 53.8 | 61.3 | 85.80 |
| Linear | 8.7 | 41.5 | 20.6 | 19.2 | 11.3 | 22.3 | 8.6 | 18.9 | 76.5 | 68.6 | 16.6 | 53.2 | 53.7 | 33.6 | 32.5 | 23.0 | 51.1 | 13.0 | 9.9 | 8.5 | 17.9 | 23.7 | 28.2 | 0.04 |
| Bias [37] | 22.4 | 82.6 | 49.7 | 66.2 | 67.7 | 69.0 | 24.3 | 54.6 | 78.7 | 91.4 | 60.0 | 72.6 | 75.7 | 65.9 | 51.0 | 35.0 | 69.1 | 70.8 | 37.6 | 21.5 | 30.7 | 47.7 | 56.1 | 0.14 |
| Adapter [7] | 35.1 | 85.0 | 56.5 | 66.6 | 71.3 | 45.0 | 24.8 | 54.9 | 76.9 | 87.1 | 63.5 | 73.3 | 75.2 | 43.8 | 49.5 | 31.2 | 61.7 | 59.3 | 23.3 | 13.6 | 29.6 | 39.0 | 52.5 | 0.76 |
| VPT-Shallow [6] | 21.9 | 76.2 | 54.7 | 58.0 | 41.3 | 16.1 | 15.1 | 40.0 | 74.0 | 69.5 | 58.9 | 72.7 | 68.8 | 40.3 | 44.7 | 27.9 | 60.5 | 11.8 | 11.0 | 12.4 | 16.3 | 28.1 | 41.2 | 0.04 |
| VPT-Deep [6] | 8.2 | 55.2 | 58.0 | 39.3 | 45.2 | 19.4 | 21.9 | 35.3 | 77.9 | 91.0 | 45.4 | 73.6 | 72.0 | 39.0 | 40.9 | 30.6 | 53.9 | 21.0 | 12.1 | 11.0 | 14.9 | 27.9 | 39.9 | 0.06 |
| LoRA [24] | 31.8 | 88.4 | 59.9 | 81.7 | 85.3 | 90.3 | 23.7 | 65.9 | 84.2 | 92.5 | 76.2 | 75.4 | 82.1 | 85.9 | 64.1 | 49.4 | 82.8 | 83.9 | 51.8 | 34.6 | 41.3 | 61.7 | 67.5 | 0.30 |
| ARC$_{att}$ | 34.8 | 89.3 | 62.0 | 85.9 | 84.4 | 91.1 | 24.8 | 67.4 | 85.8 | 93.5 | 81.3 | 75.6 | 84.1 | 84.0 | 63.5 | 51.2 | 83.0 | 89.1 | 54.0 | 34.2 | 43.0 | 62.7 | 69.0 | 0.09 |
| ARC | 31.3 | 89.3 | 61.2 | 85.9 | 83.1 | 91.6 | 24.4 | 66.7 | 86.0 | 94.0 | 80.4 | 74.8 | 83.8 | 85.8 | 64.6 | 50.5 | 82.8 | 82.8 | 53.5 | 36.3 | 39.7 | 62.0 | 68.3 | 0.13 |

Table 18: Performance comparison on VTAB-1k using Moco V3 self-supervised pre-trained ViT-Base as backbone.

| Method \ Dataset | Natural | | | | | | | | Specialized | | | | | Structured | | | | | | | | | Mean Total | Params.(M) |
|---|---|---|---|---|---|---|---|---|---|---|---|---|---|---|---|---|---|---|---|---|---|---|---|---|
| | CIFAR-100 | Caltech101 | DTD | Flowers102 | Pets | SVHN | Sun397 | Mean | Camelyon | EuroSAT | Resisc45 | Retinopathy | Mean | Clevr-Count | Clevr-Dist | DMLab | KITTI-Dist | dSpr-Loc | dSpr-Ori | sNORB-Azim | sNORB-Ele | Mean | | |
| Full fine tuning | 57.6 | 91.0 | 64.6 | 91.5 | 79.9 | 89.8 | 29.1 | 72.0 | 85.1 | 96.4 | 83.1 | 74.3 | 84.7 | 55.1 | 56.9 | 44.7 | 77.9 | 63.8 | 49.0 | 31.5 | 36.9 | 52.0 | 66.2 | 85.69 |
| Linear | 62.9 | 85.1 | 68.8 | 87.0 | 85.8 | 41.8 | 40.9 | 67.5 | 80.3 | 93.6 | 77.9 | 72.6 | 81.1 | 42.3 | 34.8 | 36.4 | 59.2 | 10.1 | 22.7 | 12.6 | 24.7 | 30.3 | 54.7 | 0.04 |
| Bias [37] | 65.5 | 89.2 | 62.9 | 88.9 | 80.5 | 82.7 | 40.5 | 72.9 | 80.9 | 95.2 | 77.7 | 70.8 | 81.1 | 71.4 | 59.4 | 39.8 | 77.4 | 70.2 | 49.0 | 17.5 | 42.8 | 53.4 | 66.4 | 0.14 |
| Adapter [7] | 73.0 | 88.2 | 69.3 | 90.7 | 87.4 | 69.9 | 40.9 | 74.2 | 82.4 | 93.4 | 80.5 | 74.3 | 82.7 | 55.6 | 56.1 | 39.1 | 73.9 | 60.5 | 40.2 | 19.0 | 37.1 | 47.7 | 64.8 | 0.98 |
| VPT-Shallow [6] | 68.3 | 86.8 | 69.7 | 90.0 | 59.7 | 56.9 | 39.9 | 67.3 | 81.7 | 94.7 | 78.9 | 73.8 | 82.3 | 34.3 | 56.8 | 40.6 | 49.1 | 40.4 | 31.8 | 13.1 | 34.4 | 37.6 | 57.9 | 0.05 |
| VPT-Deep [6] | 70.1 | 88.3 | 65.9 | 88.4 | 85.6 | 57.8 | 35.7 | 70.3 | 83.1 | 93.9 | 81.2 | 74.0 | 83.0 | 48.5 | 55.8 | 37.2 | 64.6 | 52.3 | 26.5 | 19.4 | 34.8 | 42.4 | 61.2 | 0.05 |
| LoRA [24] | 58.8 | 90.8 | 66.0 | 91.8 | 88.1 | 87.6 | 40.6 | 74.8 | 86.4 | 95.3 | 84.3 | 75.5 | 85.1 | 51.3 | 64.6 | 51.3 | 81.9 | 83.2 | 47.5 | 32.4 | 47.3 | 61.4 | 71.3 | 0.30 |
| ARC$_{att}$ | 59.3 | 90.9 | 67.7 | 93.6 | 89.2 | 90.5 | 40.3 | 75.9 | 87.1 | 94.8 | 85.4 | 75.5 | 85.7 | 84.0 | 64.9 | 52.5 | 83.1 | 88.2 | 53.4 | 33.0 | 46.2 | 63.2 | 72.6 | 0.09 |
| ARC | 60.0 | 91.3 | 67.9 | 92.8 | 89.3 | 91.4 | 40.9 | 76.2 | 87.5 | 95.6 | 86.1 | 75.6 | 86.2 | 83.0 | 64.2 | 50.2 | 80.6 | 85.0 | 53.0 | 34.6 | 47.4 | 62.3 | 72.4 | 0.13 |

# G    Broader impacts

**Efficient usability.**    Unlike previous approaches, our method incorporates a parameter sharing scheme across different layers of the model, resulting in a significant reduction in the number of parameters that need to be fine-tuned. This approach allows us to maintain competitive performance while achieving parameter efficiency. By maximizing the utilization of large-scale pre-trained models, our ARC methods offer enhanced usability and practicality in various applications.

**Environmental-friendly consumption.**    In addition to the reduction in computational overheads, another significant benefit of our method is the positive impact on carbon emissions reduction and environmental protection. By optimizing the computational efficiency of the model, we minimize the energy consumption required during the training and deployment of the model. This reduction in energy consumption leads to a decrease in carbon emissions, contributing to environmental sustainability. Our method not only delivers improved performance and efficiency but also aligns with the larger goal of mitigating the environmental impact of AI technologies.

**Ethical Considerations.**    Our model focuses on utilizing the representation and generalization capacity obtained from large-scale pre-trained datasets and models. However, it is crucial to acknowledge that if the pre-training datasets contain bias or illegal information, there is a risk of inheriting such issues into our model.

In order to address this concern, it becomes imperative to explore research directions that aim to identify and prevent privacy leakage and correct model bias. This involves developing robust mechanisms to detect and mitigate bias in training data, as well as implementing privacy-preserving techniques to safeguard sensitive information.

