# OpenReview forum: "Efficient Adaptation of Large Vision Transformer via Adapter Re-Composing"
_NeurIPS.cc/2023/Conference — NeurIPS 2023 poster_

### Official Review · Reviewer_jreR · 2023-07-02

**Soundness:** 2 fair
**Presentation:** 2 fair
**Contribution:** 2 fair
**Rating:** 3
**Confidence:** 5

**Summary:**

This work presents an efficient adapter design for transfer learning of large retrained models. The key idea is to enable adapter to be shared across layers and imposing low-rank constraint, so that the overall trainable parameters can be reduced. Given the re-composed linear adapter, existing reparameterization methods can be further used to avoid extra computation cost.

**Strengths:**

- Parameter efficient transfer learning is an important approach for making large pertained model useful in many applications.

- The paper is written to be easy to read.

**Weaknesses:**

First of all, the motivation of this work is weak -- Existing adapters are already lightweight, e.g., less than 0.5% of the pretrained model, and the scope for further reducing the size of adapter is marginal. That being said, this adapter efficiency problem is not significant.

In introduction, it is hard to read out the novelty of this method since more related works such LoRA is not discussed and compared at all.

Limited novelty with the proposed method: Low-rank constraint is not new, as already used in LoRA [24] although not in the same way, and parameter sharing across layers is also not novel idea. This work combines the two in a single place, which could be considered as being not significant.

Limited performance gain:
- In most cases, the proposed method cannot achieve a good margin over previous methods (e.g., SSF).

- Another important metric, FLOPs, would be useful to report.






**Questions:**

See the weaknesses above.

---

> ### Author Rebuttal · Authors · 2023-08-09
>
> We greatly appreciate your dedicated efforts in reviewing our work. While we value your contribution, we would like to address a few factual errors that have come to our attention. Your understanding and collaboration in rectifying these inaccuracies would be immensely valuable to us.
>
> **1. Regarding this adapter efficiency problem is not significant.**
> **Re:** Our response to this particular concern is available in Joint Response 1. To facilitate access to our reply, we are reproducing it here. We acknowledge the valid point regarding the limitations on reducing the absolute count of adaptation parameters. In light of this, we are revising our tone to convey a more balanced perspective on parameter reduction. It is essential, however, to underscore that our approach demonstrates improved adaptation performance even when operating with a reduced parameter count. It is important to note that our contribution extends beyond this efficiency aspect, as we introduce innovative insights into the domain of low-rank adaptation strategies.
>
> **2. Regarding it is hard to read out the novelty of this method since more related works such LoRA is not discussed and compared at all.**
> **Re:** We respectfully disagree with this assertion. Within our introduction, we thoroughly reviewed and analyzed pertinent parameter-efficient pre-trained adaptation methods, as demonstrated in lines 32 to 41. Notably, references [7,8] employed low-rank adapters within the context of vision applications. Additionally, we accentuate the key distinctions that set our approach apart from these existing solutions, elaborated upon from lines 42 to 48. Moreover, to provide a more comprehensive understanding of these distinctions, we offer a visual summary of typical parameter-efficient pre-trained model adaptation techniques in Fig. 1 of the original paper. This illustration is complemented by an in-depth analysis of existing methodologies. Given these comprehensive efforts, we respectfully contend that the criticism regarding the perceived novelty of our work is unfounded.
>
> **3. Regarding concerns about limited novelty with the proposed method: Low-rank constraint is not new, as already used in LoRA [24] although not in the same way**.
> **Re:** We want to clarify that our assertion has never been to claim the low-rank design of adapters as our contribution. In fact, we explicitly state our approach's novelty from line 43 to line 48 of the Introduction, where we outline, "We adopt a low-rank design for the adapter using a bottleneck operation but propose a novel approach. Unlike other methods that place the adapter in different layers and directly learn different parameters for each adapter to cater to layer-wise variation, we propose sharing the down/up projections in the low-rank adapter across different layers and simply learning low-dimensional re-scaling coefficients to re-compose the linear projections into layer-adaptive adapters." This clarifies that our innovation indeed lies in providing a fresh perspective within the domain of low-rank adaptation strategies. While the low-rank constraint itself might not be new, our distinctive approach and insights set our method apart from existing ones, such as LoRA [24].
>
> **4. Regarding concerns about limited novelty with the proposed method: parameter sharing across layers is also not novel idea**.
> **Re:** We respectfully disagree with the characterization that parameter sharing across layers lacks novelty. While the concept of parameter sharing is indeed a broad one, the crux lies in how this sharing is executed and whether it contributes fresh perspectives to the field. In our case, we delve into adaptation matrix sharing within the context of the low-rank design of adapters, specific to parameter-efficient pre-trained adaptation. As far as we are aware, our work is the inaugural exploration of this approach, imbuing the community with novel insights into the realm of low-rank-based adaptation methods. The uniqueness lies not in the general notion of parameter sharing, but in the inventive application and implications of this concept within our proposed framework.
>
> **5. Regarding limited gains: In most cases, the proposed method cannot achieve a good margin over previous methods (e.g., SSF).**
> **Re:** We appreciate your perspective on this matter. In the original paper, we executed our method with standard data augmentations and additionally re-implemented SSF under the same augmentation conditions. The findings, as detailed in Tables 1 and 2, reveal that our ARC method exhibits an improvement of approximately 3% over SSF – a substantial enhancement. Furthermore, in response to input from other reviewers, we proceeded to retrain our ARC method while integrating the data augmentation strategies outlined in the original SSF paper [9]. The outcomes, meticulously presented in Table 1 of the provided rebuttal_pdf, consistently underscore our method's superior performance compared to SSF across various datasets. These results affirm the efficacy of our approach and its propensity to yield consistent advancements over previous methodologies.
>
> **6. Regarding FLOPs**.
> **Re:** The majority of FLOPs emanate from our ARC modules, which are meticulously designed with a bottleneck structure. Consequently, the additional FLOPs introduced by our ARC approach align closely with those of LoRA and Adapter methods. Moreover, it's noteworthy that our supplementary adapter modules entail a linear mapping, enabling us to re-parameterize the ARC modules within the original pre-trained model framework without incurring additional FLOPs overhead. A detailed discussion on this aspect can be found in lines 170-176 of our paper.

---

### Official Review · Reviewer_4kbb · 2023-07-04

**Soundness:** 3 good
**Presentation:** 3 good
**Contribution:** 3 good
**Rating:** 5
**Confidence:** 4

**Summary:**

This paper introduces a parameter-efficient transfer learning method named Adapter Re-Composing (ARC), which mainly focuses on investigating the reusability of adapted parameters. The authors propose to apply a shared adapter to all the layers (blocks) of the pre-trained model, and they use different Re-Scaling Coefficients (diagonal matrices) in different layers to ensure the diversity of parameters in different layers.
The motivation behind this design is the adapter module's low-rank property, shown in Fig.3 in the main text.
Moreover, the authors conduct extensive experiments to demonstrate the effectiveness of their designs, where they train fewer parameters and achieve competitive or even better performance compared to prior arts.

**Strengths:**

1: This paper is well-written and well-organized.

2: The ARC design in the paper is well-motivated.

3: Although the technic is simple and easy to implement, the performance is impressive.

4: The experiments are comprehensive.

**Weaknesses:**

1: The ARC leverages learnable re-scaling coefficients in different layers to maintain diversity. However, the authors didn't disscuss the numerical difference among the re-scaling coefficients across different layers. If they are also similar, a share-weight adapter can replace the ARC.

2: In Tab. 1 and Tab. 2, the SSF* applies additional techniques (e.g., data augmentation) during training and gains non-trivial improvements. Why did the authors not use those techniques to improve the performance of ARC further? Will these techniques further benefit the ARC?

3: I wonder about the computational overhead of the ARC, e.g., the GPU memory usage during training, because the number of learnable parameters may not necessarily be positively correlated with the GPU memory usage. Could authors provide the additional GPU memory usage comparison between ARC and other related works?

4: How to use ARC in Hierarchical Vision Transformers such as Swin Transformer is unclear in Lines 277-279. Adding more details here may be helpful.

Overall, my major concern is weakness 1. Happy to raise my rating if my concers are well adressed.

**Questions:**

See the weakness.

**Limitations:**

The authors disscuss the limitations and boarder impact.

---

> ### Author Rebuttal · Authors · 2023-08-09
>
> We value the positive feedback you have shared and address the raised concerns as follows:
>
> **1. Regarding the diversity of re-scaling coefficients.**
> **Re:** We appreciate the thoughtful insights you've provided. To comprehensively address this matter, we conducted an extensive analysis of the re-scaling coefficients across various layers. The outcomes, specific to the DTD dataset within the VTAB-1k dataset, have been visually captured in Fig. 2 of the provided rebuttal_pdf. This analysis reveals substantial dissimilarity among the scaling factors across different layers. This outcome unequivocally underscores the distinctiveness of our proposed ARC structure, making it non-trivial to substitute with a shared adapter. Furthermore, we conducted the experiment you suggested by sharing adaptation matrices while excluding re-scaling parameters. The corresponding results are presented as "Sharing-Adapter" in Table 1 of the provided rebuttal_pdf. The considerable decline in performance observed in this scenario serves as compelling evidence attesting to the indispensable role played by the re-scaling coefficients. This exploration reaffirms the crucial significance attributed to the re-scaling mechanism in our approach.
>
> **2. Regarding run the paper's method under SSF's data augmentation**.
> **Re:** In response to your insightful suggestion, we have rigorously re-trained our ARC method while implementing the data augmentations featured in SSF [9]. The results of this endeavor have been presented in Table 1 of the provided rebuttal_pdf, and a comprehensive explanation can be found in Joint Response 2. Notably, our ARC method consistently exhibits improved performance compared to SSF [9] when subjected to the same set of augmentations, reaffirming the efficacy and generalizability of our approach.
>
> **3. Regarding GPU usage**.
> **Re:** The utilization of GPU resources during model training is influenced by multiple factors, including the model size, intermediate variables generated during the forward process, and gradient information in back propagation. Notably, intermediate variables tend to consume a substantial portion of GPU resources, surpassing the model size in significance. In alignment with the LoRA and Adapter designs, our method fine-tunes the low-rank bottleneck structure within each layer. Consequently, the accumulation of additional intermediate variables during training is akin to these methods, resulting in similar GPU memory usage for our ARC approach. In the case of SSF, a concise breakdown of the GPU usage for intermediate variables can be approximated as O(L\*m\*N\*D), where L signifies the number of layers, m denotes the count of SSF modules embedded in each layer, N represents the token count, and D stands for feature dimensions. On the contrary, the GPU usage associated with intermediate variables produced by our ARC adheres to O(L\*N\*D), markedly smaller than the SSF approach. To validate this analysis, we conducted experiments, training different models under identical hardware conditions and gauging GPU memory consumption. The outcomes confirm that the GPU utilization of our method closely aligns with the Adapter and LoRA methods, notably outperforming the SSF method in terms of resource efficiency.
>
> **4. Regarding clarification of how to use ARC in Hierarchical Vision Transformers such as Swin Transformer**.
> **Re**: We regret any confusion caused by our previous description. To provide further clarity, it's important to note that the Swin Transformer comprises distinct stages, each characterized by varying feature dimensions. Achieving global sharing across these stages is unfeasible. In Swin Transformer, each stage consists of multiple transformer blocks with uniform feature dimensions. In light of this architecture, we've introduced shared adaptation matrices between blocks within the same stage.

---

> > ### Comment · Reviewer_4kbb · 2023-08-14
> >
> > Thanks for the comprehensive reply. I read the rebuttal and all other reviews.
> >
> > My major concern is that the proposed framework is an incremental modification of the LoRA.
> >
> > First, the low-rank optimization isn't novel, given the existence of LoRA (the authors also admit that this operation isn't claimed as one of the contributions).
> >
> > Second, the "sharing" operation is very sensitive to its placed position, making the proposed method a little bit trivial. From the supplied Table 1 in the pdf, "sharing adapter across all layers" results are significantly worse than "sharing adapter across layer."
> >
> > Based on the above two facts, the ARC's contribution is limited to the "sharing down-/up-projections," which sounds a little tricky given the current results. Suppose the authors would like to claim the "sharing operation" as the main contribution. In that case, the paper should pay more attention to what makes this operation work and develop a more in-depth analysis (e.g., which layers share projections or the effect of residual connections, etc.).
> >
> > To this end, I decide to downgrade both my score and confidence to (5,4). I'm still waiting for further replies from the authors and would like to hear the opinions of other reviewers on the rebuttal.

---

> > > ### Author Response · Authors · 2023-08-14
> > >
> > > Dear Reviewer 4kbb,
> > >
> > > We are grateful for your prompt response to our work. However, ***we must address a misunderstanding that has arisen from your feedback.***
> > >
> > > In response to your original comment, "discuss the numerical difference among the re-scaling coefficients across different layers," we have taken a comprehensive approach to address this concern. We have presented a detailed analysis of the disparity among the re-scaling coefficients across layers, as evidenced by the findings showcased in Figure 2 of the attached rebuttal PDF. It is clear from this analysis that the re-scaling coefficients do indeed exhibit variations, which underscores the rationale behind our decision to learn layer-specific re-scaling coefficients. In our efforts to address your query more explicitly, we conducted experiments wherein the re-scaling coefficients were removed. The results of these experiments are meticulously documented as "Sharing-Adapter" in Table 1 of the rebuttal PDF. As anticipated, these results demonstrated a significant decline in performance. This outcome validates our assertion that the absence of layer-specific re-scaling coefficients fails to account for the nuanced layer-wise variations, leading to a lack of sufficient model capacity to adapt effectively to downstream tasks.
> > >
> > > Given these clarifications, it is crucial to emphasize that the contribution of ARC extends beyond merely "sharing down-/up-projections." The true innovation lies in the strategic acquisition of layer-specific adapters—a process that efficiently utilizes resources while delivering significant impact. While we do indeed utilize the concept of shared down-/up-projections, the heart of our innovation revolves around the careful integration of these components through the use of layer-specific re-scaling coefficients. ***It is important to understand that the re-composed adapters should not be mistakenly interpreted as a simple "sharing adapter across layer," as suggested in your feedback. Our approach involves a thoughtful arrangement that optimizes model capacity and performance.***
> > >
> > > We trust that this clarification better communicates the essence of our work and the distinctiveness of our contribution. We remain committed to addressing any further inquiries or uncertainties you may have.
> > >
> > > Thank you for your thoughtful consideration.

---

> > > > ### Comment · Reviewer_4kbb · 2023-08-15
> > > >
> > > > Thanks for the further clarification. I have read LoRA, the rebuttal, and the paper again to check the mentioned details. I tend to accept this paper but keep my rating to borderline accept.

---

> > > > > ### Author Response · Authors · 2023-08-15
> > > > >
> > > > > We would like to express our gratitude for your time and effort in thoroughly reviewing the related work, our paper, and the rebuttal. Your willingness to accept the paper is greatly appreciated.

---

### Official Review · Reviewer_4jX4 · 2023-07-07

**Soundness:** 2 fair
**Presentation:** 4 excellent
**Contribution:** 2 fair
**Rating:** 5
**Confidence:** 4

**Summary:**

The paper explores ARC, which is a novel parameter-efficient fine-tuning method which uses a similar architecture as adapters but introduces inter- and intra- layer weight sharing. Some down- and up- projection weights are shared but every adapter position uses an independent set of per-channel scaling factor on the channel-reduced intermediate features. The proposed method obtains competitive results on various vision transformer adaptation benchmarks in terms of recognition performance and number of parameters. Moreover it can be re-parameterized into adjacent fully-connected layers so no overhead is incurred during inference.

**Strengths:**

* The paper is generally well written. The figures are clear and the text is easy to follow.

* The experiments are comprehensive and cover a wide range of visual recognition datasets and the results are competitive.

**Weaknesses:**

* The effectiveness of the proposed method is not well justified. Section 3.3 does not make much sense to me: Having long-tailed singular values only implies that the weight difference can be approximated well by a low-rank matrix and thus justifies the bottleneck design. Shared projections further requires the adapters to have largely overlapped kernel and image spaces, which, instead of the singular values themselves, are determined by the direction of the singular vectors corresponding to the top singular values. It also doesn't theoretically justify the intra-block sharing design (i.e., using the transpose of down-projection as up-projection).

* In multiple places (e.g., caption of Table 1 and Line 229), the paper's claim of using simple augmentations on baselines for 'fair comparison' is questionable. Due to the vast differences in nature of different methods, it is expected that their optimal training configurations are different, and each method, including this paper's own, should have the right to choose its optimal training configuration as long as it does not violate some principal rules of machine learning (e.g., leak of test data): Conversely, it is also improper to request that the paper's method being run under SSF's data augmentation for 'fair comparison'.

**Questions:**

* The paper cited [7] in multiple tables for visual adapter results but [7] seems to not include any results on computer vision tasks (actually the paper title is *Parameter-efficient transfer learning for **NLP***). Are those results reproduced by the authors using the same architecture as [7]? If so, could the authors point to a place where more detailed settings (e.g., the bottleneck dimension, the activation function, the training hyper-parameters) of this baseline can be found?

* In Table 3, it is strange to me why the proposed methods obtain lower performance on ViT-Huge than ViT-Large. Is it possible that the proposed method or some baselines are over-fitting in extremely large backbones?

* Following weakness 1, an understanding of why the proposed method works fairly well can also be provided by experiments: For example, are features from adjacent layers similar due to the identity connections? What if the projections are progressively shared in groups of increasing sizes among consecutive layers before reaching the global sharing setting as reported in the paper?

**Limitations:**

There are no unmentioned limitations in the paper to my mind.

---

> ### Author Rebuttal · Authors · 2023-08-09
>
> We appreciate the constructive comments. We address the concerns as follows:
>
> **1. Regarding the motivation and justification of shared adaptation matrices in weakness 1**.
> **Re:** We appreciate your feedback. In response, we have incorporated the suggested analysis to provide a robust rationale for the motivation behind our proposed adaptation matrix sharing strategy. For a comprehensive understanding of this matter, we invite you to refer to Joint Response 1, where you will find a detailed elaboration on this issue.
>
> **2. Regarding run the paper's method under SSF's data augmentation for 'fair comparison' in weakness 2**.
> **Re:** In response to your insightful suggestion, we have rigorously re-trained our ARC method while implementing the data augmentations featured in SSF [9]. The results of this endeavor have been presented in Table 1 of the provided rebuttal_pdf, and a comprehensive explanation can be found in Joint Response 2. Notably, our ARC method consistently exhibits improved performance compared to SSF [9] when subjected to the same set of augmentations, reaffirming the efficacy and generalizability of our approach.
>
> **3. Regarding the results of [7] in question 1**.
> **Re:** We wish to clarify that we did not independently replicate the results from the literature [7] in the realm of computer vision. Instead, we directly adopted the experimental outcomes presented in the VPT paper [6]. For precise details pertaining to the settings employed, we kindly direct your attention to the VPT paper [6].
>
> **4. Regarding inferior performance of ViT-Huge comparing to ViT-Large**.
> **Re:** We wish to highlight that upon a thorough comparison between Table 3(a) and Table 3(b) in our paper, it becomes evident that a number of methods, including Full Fine-tuning, Adapter, VPT-Deep, and LoRA, exhibit inferior performance on ViT-Huge compared to ViT-Large. It is noteworthy that these methods possess trainable parameters that increase in correspondence with the number of layers, potentially contributing to overfitting stemming from model expansion. This phenomenon aligns with the observations delineated in the Visual Prompt Tuning (VPT) paper and harmonizes with the outcomes depicted in Fig. 4 of the VPT paper. Conversely, methods such as VPT-Shallow, which feature learnable parameters independent of layer count, or methods with a modest count of learnable parameters (e.g., Bias), tend to display less pronounced manifestations of this phenomenon.
>
> **5. Regarding question 3**.
> **Re:** We appreciate your suggestions. In response, we conducted a thorough investigation into the input features of adjacent layers within the network. Our findings indicate that there are no noteworthy similarities in this regard. It's worth noting that the visualization and analysis elucidating the correlation between adaptation matrices across layers, as presented in Joint Response 1, sufficiently addresses the concerns surrounding the justification of adaptation matrix sharing.

---

> > ### Comment · Reviewer_4jX4 · 2023-08-21
> >
> > Thanks for the responses from the authors and the other reviewers.
> >
> > The new experimental results (the proposed method using SSF's data augmentation) seem quite strong to me, which is the main reason why I'm raising the rating to a positive one.
> >
> > The drawback, however, is that the analysis of why the proposed method works effectively is still somewhat limited. For the rebuttal pdf, it is not very clear to me how the correlation between two sets of (singular) vectors are defined, and what is the reference correlation value of two random matrices. More generally, it still looks to me that the experimental or theoretical analysis part can be systematically enhanced (e.g., covering a range of tasks from easy to difficult and observing more fine-grained results in a series of progressively sharing settings) which might be too much to cover in a rebuttal phase.
> >
> > Based on the views above I'm raising the rating to a borderline accept.

---

### Official Review · Reviewer_zRDU · 2023-07-08

**Soundness:** 3 good
**Presentation:** 3 good
**Contribution:** 3 good
**Rating:** 5
**Confidence:** 4

**Summary:**

This paper proposes to further reduce the parameters of the adapter by introducing a weight-sharing scheme between different layers. To accommodate the variations across different layers, re-scaling coefficients are learned to re-compose the layer-adaptive adaptation matrices. Experiments are conducted on 24 downstream image classification tasks using various Vision Transformer variants.

**Strengths:**

+ The observation that “learned adaptation matrices naturally exhibit low-rank characteristics” as shown in Fig. 3 is quite interesting, making the motivation for the weight-sharing design compelling.
+ This paper is well-written and easy to follow. The figures are well-prepared and illustrate the core idea clearly.
+ The experiments are extensive.


**Weaknesses:**

- The comparisons to SSF shown in Tables 1 and 2 are re-implemented by the authors and data augmentations are removed. Their original performances of SSF are much higher than the proposed approach. It would be better if the authors reported the performance of the proposed approach with these advanced data augmentations.
- Fig. 3 shows the singular value distribution of the original MHA and FFN adapter. It would be better to show how the proposed approach alleviates such a problem by visualizing the value distribution after using the proposed approach.
- Though the observation of “learned adaptation matrices naturally exhibit low-rank characteristics” and the proposed weight-sharing scheme is very interesting, the motivation to further reduce the parameters of the adapter is not very compelling as the parameters of the adapter are already relatively small.


**Questions:**

Please see the weaknesses

**Limitations:**

Yes.

---

> ### Author Rebuttal · Authors · 2023-08-09
>
> Thank you for your diligent review of our paper and your acknowledgment of the strengths within our work. We have taken your feedback seriously and have formulated a detailed response to the specific issues you raised, which we present as follows:
>
> **1. Regarding incorporating advanced data augmentations in SSF.**
> **Re:** In response to your insightful suggestion, we have rigorously re-trained our ARC method while implementing the data augmentations featured in SSF [9]. The results of this endeavor have been presented in Table 1 of the provided rebuttal_pdf, and a comprehensive explanation can be found in Joint Response 2. Notably, our ARC method consistently exhibits improved performance compared to SSF [9] when subjected to the same set of augmentations, reaffirming the efficacy and generalizability of our approach.
>
> **2. Regarding how the proposed approach alleviates a problem shown in Fig. 3.**
> **Re:** We greatly appreciate your attention to this aspect. In Fig. 3, we offer a compelling insight into the remarkable low-rank attributes exhibited by the adaptation matrices. This distinctive observation underscores the feasibility of employing a shared basis for the reconstruction of these matrices. Drawing from this pivotal discovery, we introduce our ARC method, which not only shares down- and up-projection matrices across all layers but also learns low-dimensional re-calibration coefficients to reconstruct the adapters. This approach contributes to enhanced parameter efficiency. To further substantiate the rationale behind our parameter-sharing strategy, we provide a visual representation of the correlation between adaptation matrices across layers in Fig. 1 of the provided rebuttal_pdf. This visual analysis underscores a robust correlation among the adaptation matrices spanning layers, thus solidifying the justification for our parameter-sharing paradigm.
>
> **3. Regarding the motivation to further reduce the parameters of the adapter is not very compelling as the parameters of the adapter are already relatively small.**
> **Re:** Your insightful comment is greatly appreciated. We have provided a comprehensive response to this query in Joint Response 3. Thank you for raising this pertinent issue.

---

> > ### Comment · Reviewer_zRDU · 2023-08-21
> > **Thanks for the rebuttal**
> >
> > I appreciate the rebuttal and the clarification. The responses address most of my concerns. Therefore, I would retain my score.

---

> > > ### Author Response · Authors · 2023-08-21
> > > **Thank  you for your review and comments**
> > >
> > > Dear Reviewer,
> > >
> > > Thank you for your feedback and for taking the time to review our rebuttal. We're pleased to hear that the responses have effectively addressed your concerns.

---

### Official Review · Reviewer_7yx9 · 2023-07-18

**Soundness:** 3 good
**Presentation:** 3 good
**Contribution:** 3 good
**Rating:** 6
**Confidence:** 4

**Summary:**

This paper introduces a novel parameter-efficient fine-tuning method called Adapter Re-Composing (ARC). ARC effectively reuses parameters across different layers, resulting in remarkable improvements in performance across 24 image classification datasets while utilizing fewer learnable parameters.
The experimental evaluation conducted on various downstream datasets provides compelling evidence of ARC's superiority. It outperforms existing methods and establishes a new benchmark for parameter-efficient fine-tuning techniques.


**Strengths:**

1. The proposed ARC method is simple yet effective, achieving superior performance on multiple downstream datasets while utilizing fewer trainable parameters.

2. The experiments conducted provide compelling evidence, as they encompass various datasets, attention-based architectures, and ablations, ensuring the robustness and reliability of the findings.


**Weaknesses:**

1. In Table 3, it is observed that the performance of ViT-Huge is lower than ViT-Large. It would be beneficial if the authors could provide an explanation for this disparity.

2. The utilization of symmetric matrices ($W_{up} = W_{down}^T$) in the bottleneck design helps reduce the number of learnable parameters. However, it would be interesting to explore whether further improvements in performance can be achieved by making the downsampling and upsampling matrices independent. It would be valuable if the authors could provide a comparison of performance and parameter statistics to address this potential enhancement.

3. The experiments conducted have demonstrated the effectiveness of the proposed method. However, it would greatly enhance the strength of this paper if the authors could supplement these empirical results with theoretical analysis.

**Questions:**

Please see Weaknesses.

**Limitations:**

Please see Weaknesses.

---

> ### Author Rebuttal · Authors · 2023-08-09
>
> Thank you for recognition of our work strength. We extend our sincere appreciation for the invaluable guidance you have provided. We respond to your concerns as follows.
>
> **1. Regarding the performance of ViT-Huge is lower than ViT-Large.**
> **Re:** We wish to highlight that upon a thorough comparison between Table 3(a) and Table 3(b) in our paper, it becomes evident that a number of methods, including Full Fine-tuning, Adapter, VPT-Deep, and LoRA, exhibit inferior performance on ViT-Huge compared to ViT-Large. It is noteworthy that these methods possess trainable parameters that increase in correspondence with the number of layers, potentially contributing to overfitting stemming from model expansion. This phenomenon aligns with the observations delineated in the Visual Prompt Tuning (VPT) paper and harmonizes with the outcomes depicted in Fig. 4 of the VPT paper. Conversely, methods such as VPT-Shallow, which feature learnable parameters independent of layer count, or methods with a modest count of learnable parameters (e.g., Bias), tend to display less pronounced manifestations of this phenomenon.
>
> **2. Regarding making the downsampling and upsampling matrices independent.**
> **Re:** To offer clarity on the matter of making the symmetric matrices in the bottleneck structure independent of each other, we direct attention to Table 5(c) within our paper. In the last row of the aforementioned table, we elucidate the outcomes stemming from this adjustment, which regrettably fail to yield discernible enhancements in performance, despite introducing an elevation in parameter count. It is important to underscore that our rationale for adopting a symmetric structure in the design of adapters is expounded upon in Joint Response 1.
>
> **3. About theoretical analysis about the method.**
> **Re:** The key innovation underpinning our paper resides in our departure from learning low-rank adapters in isolation for each layer, and instead, reconstituting these adapters by recalibrating shared down- and up-projection matrices. To fortify the rationale for adaptation matrix sharing we present a visual depiction of the interlayer correlation among adaptation matrices in Fig. 1 of the rebuttal_pdf. Precisely, we apply the Singular Value Decomposition (SVD) technique to down- and up-projection matrices within adapters of every layer, gauging the alignment of right singular vectors in down-projection matrices and left singular vectors in up-projection matrices across layers. From the discernible patterns in Fig. 1(a) and Fig. 1(b) of the rebuttal_pdf, a conspicuous correlation is evident among the low-rank adapters spanning layers, substantiating the principle of shared projection matrices within our method. Furthermore, we expound the rationale for employing transpose down-projection matrices as up-projection matrices, as demonstrated in Fig. 1(c) of the rebuttal_pdf, where the strong correlation between the right and left singular vectors within each layer underscores the justification for this approach.

---

> > ### Comment · Reviewer_7yx9 · 2023-08-21
> >
> > Dear authors,
> >
> > I have meticulously reviewed the rebuttal and taken into consideration the comments provided by the other reviewers. The majority of my concerns have been addressed, and as a result, I am inclined to maintain my initial score.

---

> > > ### Author Response · Authors · 2023-08-21
> > > **Thank you for your review and comments**
> > >
> > > Dear Reviewer,
> > >
> > > We sincerely thank you for your thorough review of our paper and for dedicating your time to assess our rebuttal and the comments from other reviewers. We are delighted to learn that the majority of your concerns have been satisfactorily addressed, and we truly appreciate your positive feedback on our work.

---

### Author Rebuttal · Authors · 2023-08-09

**This concludes the the Joint Response to all reviewers.**

We appreciate the diligent efforts undertaken by both the reviewers and the ACs in thoroughly reviewing our manuscript. It is noteworthy that a consensus has been reached among four out of the five reviewers regarding the novelty and efficacy of our proposed methods. Within this context, we diligently address the shared concerns highlighted by the reviewers as **Joint Response**, and in addition, provide individualized responses to the specific inquiries raised by each reviewer.


**1. Regarding the motivation for adaptation matrix sharing across layers.**
**Re:** The key innovation underpinning our paper resides in our departure from learning low-rank adapters in isolation for each layer, and instead, reconstituting these adapters by recalibrating shared down- and up-projection matrices. This transformative approach is partially substantiated in Fig. 3 of the original paper, where the adaptation matrices exhibit pronouncedly low-rank characteristics, facilitating the use of a common basis to reconstruct the low-rank adapters. To fortify the rationale for adaptation matrix sharing, as suggested by Reviewer 4jX4 we present a visual depiction of the interlayer correlation among adaptation matrices in Fig. 1 of the rebuttal_pdf. Precisely, we apply the Singular Value Decomposition (SVD) technique to down- and up-projection matrices within adapters of every layer, gauging the alignment of right singular vectors in down-projection matrices and left singular vectors in up-projection matrices across layers. From the discernible patterns in Fig. 1 (a) and Fig. 1 (b) of the rebuttal_pdf, a conspicuous correlation is evident among the low-rank adapters spanning layers, substantiating the principle of shared projection matrices within our method. Furthermore, we expound the rationale for employing transpose down-projection matrices as up-projection matrices, as demonstrated in Fig. 1 (c) of the rebuttal_pdf, where the strong correlation between the right and left singular vectors within each layer underscores the justification for this approach.

**2. Applying the data augmentations in SSF [9] to the proposed ARC method.**
**Re:** In our original paper, we employed standard data augmentations for the proposed method. However, recognizing the advanced data augmentations used by SSF [9], we re-implemented SSF and presented results from both the original [9] and our re-implementation. In response to the insightful suggestions of Reviewers 4jX4 and 4kbb, we undertook re-training of the proposed ARC method using the same data augmentations as in SSF [9], and subsequently showcased the outcomes across the 19 datasets of VTAB-1k. Remarkably, our results indicated that with the integration of these advanced data augmentations, the proposed ARC method surpassed SSF [9] on 14 out of the 19 datasets. This noteworthy observation provides further compelling evidence underscoring the robust applicability and efficacy of our proposed method.

**3. Regarding the motivation to further reduce the parameters of the adapter is not very compelling as the parameters of the adapter are already relatively small.**
**Re:** We acknowledge the point that the scope for reducing the absolute count of adaption parameters is constrained. Therefore, we will adopt a more measured tone concerning parameter reduction. However, it's crucial to emphasize that our approach attains enhanced adaptation performance even with a reduced parameter count. Importantly, our contribution extends to providing novel insights into the realm of low-rank adaptation strategies.

---

### Decision · Program_Chairs · 2023-09-21

**Decision:**

Accept (poster)

**Comment:**

This paper presents a new Adapter Re-Composing (ARC) method for parameter-efficient fine-tuning large models. The method is simple yet effective and achieves superior performance on multiple downstream datasets while utilizing fewer trainable parameters. After rebuttal, most of reviewers give positive scores. Reviewer jreR gives quite low score given short reviews. As authors rebut, the raised concerns mostly come from misunderstanding. The AC looks into paper details carefully, and agrees with the contribution this simple yet powerful ARC technique. Thus AC decides to accept it.